# TAGA: Text-Attributed Graph Self-Supervised Learning by Synergizing Graph and Text Mutual Transformations

## Abstract

Text-Attributed Graphs (TAGs) enhance graph structures with natural language descriptions, enabling detailed representation of data and their relationships across a broad spectrum of real-world scenarios. Despite the potential for deeper insights, existing TAG representation learning primarily omit the semantic relationship among node texts, and mostly relies on supervised methods, necessitating extensive labeled data and limiting applicability across diverse contexts. This paper introduces a new self-supervised learning framework, **T**ext-**A**nd-**G**raph Multi-View **A**lignment (**TAGA**), which overcomes these constraints by integrating TAGs' structural and semantic dimensions. TAGA constructs two complementary views: Text-of-Graph view, which organizes node texts into structured documents based on graph topology, and the Graph-of-Text view, which converts textual nodes and connections into graph data. By aligning representations from both views, TAGA captures joint textual and structural information. In addition, a novel structure-preserving random walk algorithm is proposed for efficient training on large-sized TAGs. Our framework demonstrates strong performance in zero-shot and few-shot scenarios across eight real-world datasets.

## 1 Introduction

Text-Attributed Graphs (TAGs) are text documents that are connected in graph structures, allowing for deeper analysis and interpretation of complex relationships (Zhang et al., 2024; Jin et al., 2023c;a). TAGs are prevalently used in numerous real-world applications, such as social networks (Paranyushkin, 2019), citation networks (Liu et al., 2013), and recommendation systems (Wu et al., 2022). TAGs encompass textual content in both nodes and edges that elucidate the meaning of individual documents and who they are semantically correlated with. For instance, a scientific article network is a type of TAG that stores the texts of research papers and details about how they cite, criticize, and summarize each other within paragraphs. As shown in Figure 1(a), extracting knowledge like "*the first law proposed in Paper A is a special case of Paper B's Theorem 1 when under macro scale and low velocity*" from this scientific article network requires jointly considering semantics, topology, and their entanglement in the TAG.

Representation learning on TAGs is a promising, yet open research area that starts to attract fast-increasing attention (Ye et al., 2023; Wang et al., 2024; Chen et al., 2024; Hu et al., 2023; Fatemi et al., 2023; Tang et al., 2023; Li et al., 2023). Existing TAG representation learning methods typically treat each text document as an independent node embedding and then rely entirely on message passing mechanisms to model the interaction between different texts. These approaches ignore the semantic-level textual connections between different nodes. Additionally, existing works are typically only applicable for supervised learning, which require extensively labeled data that is often unavailable in real-world scenarios. Moreover, the reliance on supervised tasks means that models are usually optimized for specific tasks and domains reflected in the training dataset, which significantly constrains their applicability to new domains or broader tasks. This limitation undermines

the unique advantage of TAGs to leverage their universal linguistic attributes effectively. Although there are some graph pre-training models (Hou et al., 2022; Veličković et al., 2018; You et al., 2020; Li et al., 2023) operate in an unsupervised manner, they often focus on either graph topology or node features independently, neglecting the crucial interplay between textual semantics and structural information inherent in TAGs.

Therefore, there is a pressing need for a method that comprehensively addresses the unique nature of TAGs, seamlessly integrating both their structural and semantic dimensions within a unified unsupervised framework. This presents a significant research challenge with several substantial hurdles to overcome. **Primarily, developing a representation that can simultaneously leverage textual semantic content, graph structure, and their complex interplay presents significant challenges.** Incorporating the collective semantic-level textual connections between individual documents plays a key role in obtaining high quality representations from TAGs. **In addition, the scarcity of labeled training data further exacerbates this issue**, making traditional supervised approaches impractical and necessitating innovative unsupervised strategies. **Furthermore, the computational demands of such representation learning are substantial.** Integrating large pre-trained language models (PLMs) for processing textual corpora in TAGs imposes a significant computational burden. How to achieve high expressive representations while keeping computational requirements low, ensuring scalability and practicality for real-world applications poses a significant challenge.

In order to address the aforementioned challenges, this paper proposes a new self-supervised learning framework named **T**ext-**A**nd-**G**raph Multi-View **A**lignment (**TAGA**). TAGA jointly preserves rich semantic information, topology information, and their interplay by aligning representations of TAGs from two complementary views: the *Text-of-Graph* view and the *Graph-of-Text* view. As illustrated in Figure 1, these two views offer different representing formats of a TAG yet contain equivalent information. Specifically, the *Text-of-Graph* view organizes node texts into a structured textual document according to the TAG's topology. We also propose a novel Graph2Text encoding module to automatically transfer a TAG to a structured textual document, which is readily to be processed by language models. Conversely, the *Graph-of-Text* view represents textual nodes and topology in graph-structured data, which is then processed by a graph representation learning

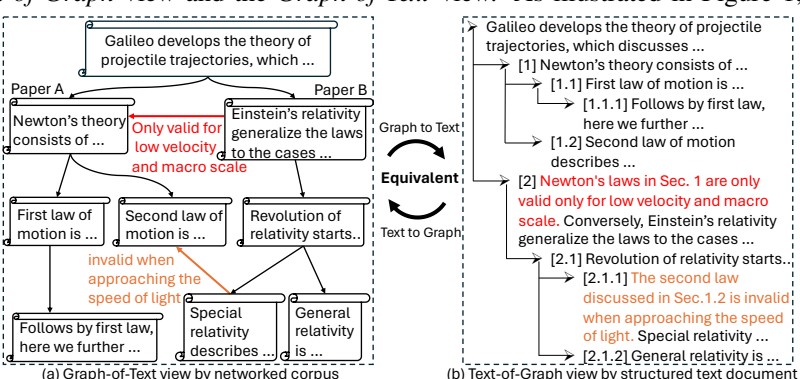

(a) Graph-of-Text view by networked corpus        (b) Text-of-Graph view by structured text document

Figure 1: Illustration of the two distinct views of TAGs: (left) Graph-of-Text and (right) Text-of-Graph. Graph-of-Text view constructs a graph-structured data over the individual text corpora, while Text-of-Graph view organizes the text node and their connection description in a hierarchical layout document. These two views can be mutually transformed to each other.

module (e.g. graph neural network). By aligning the representations learned from these two views, we encourage the learned representation to capture both textual and structural information, resulting in a unified, comprehensive representation of the TAG. Finally, to accelerate the training process, we propose a novel structure-preserving random walk algorithm.

## 2 RELATED WORKS

### 2.1 TEXT-ATTRIBUTED GRAPHS REPRESENTATION LEARNING

Existing methods typically focus on supervised learning. GraphFormers (Yang et al., 2021) introduce GNN-nested Transformers to simultaneously capture graph topology and textual semantics, enhancing

interactions between textual content and graph structure. Learning on Large-scale Text-attributed Graphs via Variational Inference (Zhao et al., 2022) presents a variational inference framework that efficiently learns node representations on large-scale TAGs. Patton (Jin et al., 2023b) pretrains language models on text-rich networks to capture semantic relationships. Recent developments have also seen efforts (Wen & Fang, 2023; Tang et al., 2023; Li et al., 2023) in aligning graph representations with textual representations. For instance, G2P2 (Wen & Fang, 2023) employs contrastive learning to align GNN representations with text encoder outputs by averaging individual node text embeddings across various neighborhood hops during its pre-training phase. However, these methods often simplify the treatment of textual encoder embeddings for neighborhoods by averaging the embeddings of individual nodes. Similarly, GRENADE (Li et al., 2023) implements a dual-level alignment strategy. This approach overlooks the underlying interactions within neighborhoods, leading to a loss of information that could be crucial for the contrastive objectives of alignment models.

## 2.2 UNSUPERVISED GRAPH PRE-TRAIN METHODS

Existing unsupervised graph pre-training methods can be categorized into several categories based on their objectives and architectures. Graph autoencoder methods, graph autoencoder methods (Kipf & Welling, 2016; Hou et al., 2022) convert node and edge features into low-dimensional embeddings, which are then used to reconstruct the original graph data. Contrastive learning approaches, like DGI (Veličković et al., 2018), GraphCL (You et al., 2020), GRACE (Zhu et al., 2020), and $S^3$-CL (Ding et al., 2023b), generate perturbed graph pairs by altering structural features, such as adding or removing nodes and edges or masking features, aiming to align the embeddings of these modified graphs closer in the embedding space. However, these methods often produce domain-specific embeddings with limited generalization ability across different domains, reducing their effectiveness in data-scarce or label-limited scenarios.

## 2.3 GRAPH2TEXT ENCODING METHODS

Recently, research include approaches (Ye et al., 2023; Wang et al., 2024; Chen et al., 2024; Hu et al., 2023; Huang et al., 2023; Fatemi et al., 2023) that first transform the TAG into text sequence and then directly utilize LLMs as the predictor given the transformed text and corresponding question as input prompt. Some methods (Tang et al., 2023; Wen & Fang, 2023) omit crucial connectivity information between nodes, while others (Fatemi et al., 2023; Huang et al., 2023) explicitly list all connections in a manner that is unnatural and difficult for language models to process.

## 3 PRELIMINARIES

A TAG can be represented as $\mathcal{G} = (\mathcal{V}, \mathcal{E}, \mathcal{C})$, where $\mathcal{V} = \{v_1, v_2, ..., v_N\}$ is a set of $N$ nodes and $\mathcal{E} \subseteq \mathcal{V} \times \mathcal{V}$ is the set of $M$ edges. $e_{ij} \in \mathcal{E}$ is an edge connecting nodes $v_i$ and $v_j \in \mathcal{V}$. $\mathcal{C} = \{C_1, C_2, \ldots, C_N\}$ is the set of node textual features where each $C_i$ is the textual corpus associated with node $v_i \in \mathcal{V}$.

The main goal of this paper is to learn the representation $f(\mathcal{G})$ of a TAG $\mathcal{G} = (\mathcal{V}, \mathcal{E}, \mathcal{C})$, which is an open research problem with several subsantial and unique challenges to be resolved. First, how the representation can jointly preserve the rich semantic information, graph information, and their interplay in TAG. Moreover, the unavailability of the training labels further troubles the representation learning. Second, the efficiency and scalability present a big challenge in representation learning of TAG because of the synergization of computational overhead of LLMs and the large corpus to be considered in the subgraph of TAG.

## 4 METHODOLOGY

To effectively address the substantial challenges of unsupervised representation learning on TAGs, we propose a novel self-supervised learning framework called **T**ext-**A**nd-**G**raph Multi-View **A**lignment (**TAGA**). Specifically, to jointly preserve both rich semantic information, topology information, and their interplay, we propose to learn and align the representations of TAG in two complementary views, namely text view and graph view. In particular, the text view is a *Text-of-Graph*, where the TAG's node texts are organized according

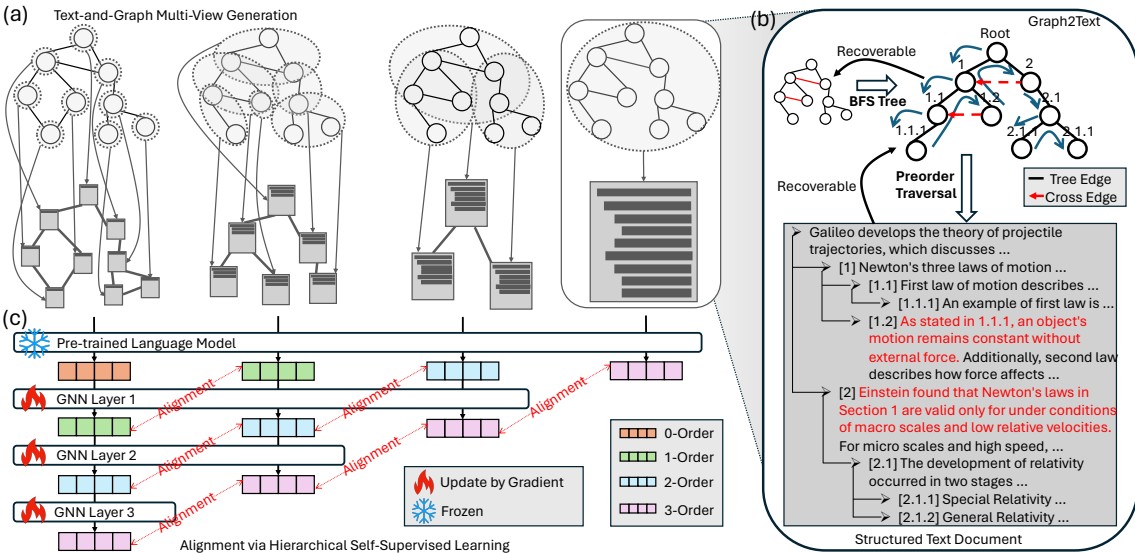

Figure 2: Illustration of the proposed self-supervised learning framework. (a) Generation of different orders of *Graph-of-Text* views; (b) The Graph2Text module that transforms a *Graph-of-Text* view into a *Graph-of-Text* view; (c) The alignment module via hierarchical self-supervised learning.

to the TAG's topology into a collective textual hierarchical document, which inherently has the power to encompass logic and relational information among different node texts. The graph view is a *Graph-of-Text*, where the TAG's nodes and topology are turned into a graph structured data. These two views contain equivalent information but in different formats, allowing them to mutually supervise each other. Then the text view can be transformed by PLMs, which are adept at preserving textual information, while the graph view can be transformed by GNN, which are designed to guarantee preserving graph information. Therefore, by aligning the representations learned from these two views, we encourage the graph view's representation to also capture textual information and the text view's representation to also capture graph information. The above new idea is shown in Figure 2, where Figure 2(a) illustrates construction of *Graph-of-Text* view while Figure 2(b) illustrates *Text-of-Graph* view, as detailed in Section 4.1. In Section 4.2, we propose the Graph2Text module that can information loselessly transform the *Graph-of-Text* view to *Text-of-Graph* view. Their respectively transformed embeddings are aligned by our proposed TAG-hierarchical self-supervised learning framework, which is elaborated in Section 4.3. Finally, a novel acceleration algorithm of our learning process to reduce computational complexity to near linear is detailed in Section 4.4.

## 4.1 TEXT-AND-GRAPH MULTI-VIEW CONSTRUCTION

Existing methods for learning representations on TAGs typically simply use GNNs to aggregate individual node embeddings generated from node texts. These methods lack the ability to consider the textual semantic relationship between different node texts in a joint document, and usually require supervised labels for training. Moreover, the resulting embeddings often lack generalization capabilities beyond the specific domain and task of their training data. To address these, our proposed framework **TAGA** first leverages two views of a TAG: *Text-of-Graph* (*TofG*) and *Graph-of-Text* (*GofT*). Each view can be defined at different neighborhood orders, allowing for a multi-order hierarchical representation. Specifically, a $k$-order *TofG* view represents a node's $k$-hop neighborhood as a single textual corpus that encompasses all nodes and their connections within that neighborhood. This corpus is then processed by a PLM to extract semantic embeddings that capture the combined content and structure within that $k$-hop neighborhood. In contrast, the corresponding $k$-order *GofT* view is constructed as a graph structure, where nodes represent lower order *TofG*s within the $k$-hop neighborhood. A GNN model is then applied to aggregate information from these connected lower order

*TofG*s, capturing the overall neighborhood context. This ensures that both *TofG* and *GofT* views at the same order encode equivalent information about the neighborhood.

To illustrate, consider a node with a 3-hop neighborhood, as shown in Figure 2(a). Its 3-order *TofG* is constructed by transforming the entire 3-hop neighborhood as a single text corpus. Three distinct 3-order *GofT* views can then be created using *TofG*s of orders 0, 1, and 2 as nodes in the graph structure. To maintain information consistency, the number of GNN aggregation layers decreases with increasing *TofG* order: 3 layers for 0-order *TofG*s, 2 for 1-order *TofG*s, and 1 for 2-order *TofG*s. This ensures that each 3-order *GofT* view captures the same 3-hop neighborhood information as the 3-order *TofG* view, facilitating information equivalent views to enable further self-supervised learning alignment.

## 4.2 REPRESENT TEXT NEIGHBORHOOD INFORMATION VIA HIERARCHICAL DOCUMENT LAYOUT

The key to our proposed self-supervised learning framework is ensuring that the two distinct graph views (*TofG* and *GofT*) contain equivalent information. This necessitates constructing a *TofG* view through the $\mathrm{Graph2Text}$ module that preserves all connectivity information present in the original TAG. Existing methods (Fatemi et al., 2023; Huang et al., 2023; Wen & Fang, 2023; Tang et al., 2023) often struggle to effectively represent the structural information of graphs in a way that is both comprehensive and natural to language model understanding. These methods typically designs text templates to explicitly describe local graph structure by stating nodes and how they are connected in plain text. For example, *"The first node is . . . . The second node is . . . . First node connects to third node. Second node connects . . . "*. However, these methods usually fails to fully leverage the pretrained capabilities of language models because they do not present the structure in a natural language-speaking manner. This discrepancy between the transformed graph text and the original pre-training corpus leads to a distributional shift, hindering the PLM's ability to generate high-quality embeddings that accurately reflect both the semantic and structural aspects of the TAG.

To address this issue, we introduce a novel $\mathrm{Graph2Text}$ approach that transforms a graph neighborhood into a hierarchical text document. This hierarchical structure mirrors the original graph's topology, ensuring that the document's latent structure is equivalent to the graph itself. Crucially, the resulting document resembles a natural document, aligning with the distribution of majority text data used to pre-train PLMs. This alignment mitigates the distributional shift issue, allowing PLMs to generate embeddings that accurately reflect both the semantic and structural aspects of the graph.

Specifically, the structure of a node and its k-hop neighborhood can be represented as an ego graph, with the node itself as the root. This ego graph can be decomposed into a hierarchical tree backbone and a set of cross-edges, as illustrated in Figure 2(b). The reading order is established for the *TofG* document through a pre-order traversal of this tree structure (first visit the root, then the left subtree, then the right subtree), capturing the hierarchical relationships between nodes. To fully represent the neighborhood's structure, we then incorporate cross-edges into the document. These cross-edges indicate connections from later sections of the document back to earlier ones, effectively mirroring the original graph's topology within the text format.

As shown in Algorithm 1, the $k$-hop neighborhood of a target node $v$ in graph $G$ is represented as an ego-graph $\mathcal{G}(v, k)$. A breadth-first search (BFS) tree $\hat{\mathcal{T}}(v, k)$, rooted at $v$, provides a hierarchical structure for the document, while cross-edges (edges outside the BFS tree) are identified. A pre-order traversal of $\hat{\mathcal{T}}(v, k)$ establishes the document's hierarchical layout, assigning each node a section number. Cross-edges are then integrated by adding references at source nodes to the sections containing their respective destination nodes, if the destination node appears earlier in the traversal. This approach ensures that the document faithfully reflects the graph's structure.

## 4.3 MULTI-VIEW ALIGNMENT VIA TAG HIERARCHICAL SELF-SUPERVISED LEARNING

Upon construction of both views at different orders, a hierarchical self-supervised learning module is proposed to align the embeddings from both views. Given a TAG $\mathcal{G}$ with at most $K$-hop neighborhood size, for each

node $v_i \in \mathcal{V}$, its $k$-hop neighborhood can be denoted as $\mathcal{N}_k(v_i)$ and its corresponding $k$-order *TofG* view embedding can be represented as:

$$\mathbf{h}_k(v_i) = \text{PLM}\left(\textit{TofG}(v_i; k)\right),$$
$$\textit{TofG}(v_i; k) = \text{Graph2Text}\left(v_i \cup \mathcal{N}(v_i, k)\right), \tag{1}$$

where PLM is a pre-trained language model (e.g. BERT (Devlin et al., 2018) or LlaMA (Touvron et al., 2023)). Graph2Text is an encoding template function that can transform individual nodes and edges text into a textual corpus. Meanwhile, its corresponding $k$-order *GofT* views embeddings can be denoted as GNN aggregated representations of lower order *TofG*s:

$$\mathbf{b}_k^l(v_i) = f^{(k-l)}\left(\{\mathbf{h}_l(v_b)|v_b \in v_i \cup \mathcal{N}(v_i, k-l)\}\right), \tag{2}$$

where $l$ covers from 0 to $k - 1$ and $f^{(k-l)}$ denotes the GNN model with $k - l$ layers.

By aggregating $k - l$ layers of information over the connected $l$-order *TofG*s, the obtained $k$-order *GofT* embeddings cover equivalent information with the $k$-order *TofG* view embedding. Therefore, given all the embeddings from level 1 to $K$, the supervision objective function can be written as:

$$\mathcal{L}_{\text{positive}} = -\frac{1}{K|\mathcal{B}|} \sum_{v_i \in \mathcal{B}} \sum_{k \in [1,K]} \sum_{l \in [0,k-1]} \rho\left(\mathbf{b}_k^l(v_i), \mathbf{h}_k(v_i)\right), \tag{3}$$

where $\mathcal{B}$ represents the minibatch and $\rho$ denotes a similarity function, such as cosine similarity. Additionally, we include the negative samples that chosen from other nodes within the minibatch:

$$\mathcal{L}_{\text{negative}} = \frac{1}{K|\mathcal{B}|} \sum_{v_i, v_j \in \mathcal{B}, v_1 \neq v_2} \sum_{k \in [1,K]} \sum_{l \in [0,k-1]} \rho\left(\mathbf{b}_k^l(v_i), \mathbf{h}_k(v_j)\right), \tag{4}$$

Thus, the overall objective function can be denoted as:

$$\mathcal{L} = \mathcal{L}_{\text{positive}} + \mathcal{L}_{\text{negative}} \tag{5}$$

**Time Complexity Analysis.** Consider a TAG with a maximum $K$-hop neighborhood size, where each node has an average degree $d$ and text attribute length $L$. Assume the feature dimensionality is $F$. In the case of transformer-based PLMs, the time complexity for processing the *TofG* view of a node would be $O((dL)^2K^2)$, due to the quadratic complexity of self-attention mechanisms with respect to input sequence length. In contrast, our method employs a GNN to aggregate information from lower-order *TofG*s, each of length $dL$. Assuming a GNN with constant complexity per layer, the time complexity for aggregating information from all $K$ levels of the *GofT* view would be $O(L^2dK)$. Our method achieves significantly higher efficiency than directly using PLMs for *TofG* views, with details available in the Appendix C.

## 4.4 ACCELERATING TRAINING ON LARGE TAGs WITH STRUCTURE-PRESERVING RANDOM WALK

While TAGA significantly improves efficiency during inference by transferring knowledge from the PLM to a GNN model, the pre-training stage still encounters computational bottlenecks due to the quadratic complexity of transformers with respect to context length when generating *TofG* view embeddings. Existing graph sampling methods (e.g., node or edge dropping) can partially alleviate this issue, but at the cost of sacrificing valuable structure information, which is crucial for capturing the intricate relationships within TAGs.

To address this issue while preserving the structure of corpus, we propose a novel approach inspired by human reading patterns. Our method segments the hierarchical corpus into multiple related sub-corpora, mirroring how humans naturally engage with complex documents: starting with a general overview (top of the hierarchy) and delving into specific sections (sub-corpora). By navigating the corpus multiple times, focusing on different sub-corpora each time, the combined insights gained can effectively approximate the understanding achieved from processing the entire corpus.

**Algorithm 1** Hierarchical Document Layout (HDL) for Graph2Text

**Input:** Graph $G$, target node $v$, hop count $k$
**Output:** Hierarchical text document $D$

1: $\hat{\mathcal{G}}(v,k) \leftarrow$ Construct ego-graph of $v$ up to $k$ hops in $G$
2: $\hat{\mathcal{T}}(v,k) \leftarrow$ BFS tree of $\hat{\mathcal{G}}(v,k)$ rooted at $v$
3: $\hat{\mathcal{E}}^{\text{cross}}(v,k) \leftarrow$ Cross-edges in $\hat{\mathcal{G}}(v,k)$
4: $D \leftarrow$ Assign document sections to nodes following pre-order traversal
5: **for** each cross-edge $e = (u,w)$ **do**
6:     **if** $w$ precedes $u$ **then**
7:         Add reference at $u$ to section containing $w$ in $D$
8:     **end if**
9: **end for**
10: **return** $D$

**Algorithm 2** Structure-Preserving Random Walk Traversal

**Input:** Root node $v$, cross-edge probability $p$, maximum length $L$
**Output:** Traversal path $P$

1: $P \leftarrow [v]$
2: **while** $|P| < L$ and $v$ has children **do**
3:     **if** random() ¡ $p$ and $v$ has cross-edges **then**
4:         $v \leftarrow$ Random neighbor by cross-edge
5:     **else**
6:         $v \leftarrow$ Random child of $v$
7:     **end if**
8:     $P \leftarrow P + [v]$
9: **end while**
10: **return** $P$

To facilitate this behavior, we introduce a random walk-based neighborhood traversal algorithm. It simulates a reader starting at the root node and progressing towards leaf nodes in the BFS tree, transitioning from general to specific information. Additionally, at each step, there is a probability $p$ of jumping to another node via cross-edges, imitating the non-linear navigation often observed in human reading (e.g., jumping to related topics or backtracking). By averaging multiple random walk traversals, the generated paths can approximate the complete corpus. As detailed in Algorithm 2, each traversal begins at the root node $v$ and iteratively samples child nodes to form a path down the hierarchy. At each step, a jump to another node via cross-edges is possible with probability $p$. This traversal continues until reaching a predefined length or a leaf node.

## 5 EXPERIMENTS

In this section, the experimental settings are introduced first in Section 5.1, then the zero-shot and few-shot node classification performances are presented in Section 5.2, and link prediction performance is presented in Appendix B.3. We further present the effectiveness under transfer learning settings in Section 5.3. We measure model efficiency in Section 5.5. The effectiveness of framework components through ablation studies is in Section 5.4. The parameter sensitivity experiments are present in Appendix B.2 due to space limit.

### 5.1 EXPERIMENTAL SETTINGS

**Datasets.** We evaluate on eight real-world text-attributed graph datasets across different domains. Specifically, three citation networks Cora (Yang et al., 2016), Pubmed (Yang et al., 2016) and Arxiv (Hu et al., 2020), two book networks Children (Shchur et al., 2018) and History (Shchur et al., 2018), and three E-commerce networks Computers (Shchur et al., 2018), Photo (Shchur et al., 2018), and Sports (Yan et al., 2023) are chosen as our evaluation datasets. Datasets statistics can be found in Table 1.

**Comparison Methods.** We choose the textual embedding of the text corpus as the baseline, which is denoted as "PLM" in our experimental results tables. Additionally, we compare our proposed framework with six state-of-the-art graph pre-train methods. Specifically, GraphMAE (Kipf & Welling, 2016) — utilizes masked autoencoder technique to predict of graph structure and node features. GraphCL (You et al., 2020) and GRACE (Zhu et al., 2020) applies various graph augmentations to generate contrastive pairs. GraphFormers (Yang et al., 2021) and Patton (Jin et al., 2023b) insert GNN layer into transformers architecture. G2P2 (Wen & Fang, 2023) aligns GNN embeddings and text encoder embeddings through contrastive learning.

**Implementation Details.** We choose two different pre-trained language models (OpenAI's `text-embedding-3-small` (OpenAI, 2023) and `UAE-Large-V1` (Li & Li, 2023)) to generate text

| $k$-Shot | Model | Arxiv | Children | Computers | Cora | History | Photo | Pubmed | Sports |
|---|---|---|---|---|---|---|---|---|---|
| | # Nodes | 169,343 | 76,875 | 87,229 | 2,708 | 41,551 | 48,362 | 19,717 | 173,055 |
| | # Edges | 1,166,243 | 1,554,578 | 721,107 | 10,556 | 358,574 | 500,939 | 44,338 | 1,773,594 |
| | Avg # Words | 220.7 | 199.3 | 90.7 | 148.2 | 218.7 | 144.5 | 50.1 | 9.8 |
| 0 | PLM | $0.500 \pm 0.001$ | $0.094 \pm 0.003$ | $0.427 \pm 0.001$ | $0.624 \pm 0.005$ | $0.169 \pm 0.001$ | $0.387 \pm 0.009$ | $0.475 \pm 0.008$ | $0.316 \pm 0.002$ |
| | GraphMAE | $0.104 \pm 0.001$ | $0.021 \pm 0.001$ | $0.049 \pm 0.001$ | $0.194 \pm 0.006$ | $0.019 \pm 0.001$ | $0.152 \pm 0.001$ | $0.438 \pm 0.001$ | $0.112 \pm 0.001$ |
| | GraphCL | $0.089 \pm 0.001$ | $0.037 \pm 0.001$ | $0.173 \pm 0.001$ | $0.176 \pm 0.003$ | $0.191 \pm 0.001$ | $0.174 \pm 0.001$ | $0.368 \pm 0.001$ | $0.140 \pm 0.001$ |
| | GRACE | $0.045 \pm 0.001$ | $0.034 \pm 0.001$ | $0.169 \pm 0.001$ | $0.146 \pm 0.004$ | $0.079 \pm 0.001$ | $0.025 \pm 0.001$ | $0.335 \pm 0.001$ | $0.057 \pm 0.001$ |
| | GraphFormers | $0.465 \pm 0.003$ | $0.076 \pm 0.001$ | $0.147 \pm 0.001$ | $0.641 \pm 0.004$ | $0.185 \pm 0.005$ | $0.192 \pm 0.003$ | $0.441 \pm 0.005$ | $0.368 \pm 0.002$ |
| | PATTON | $0.496 \pm 0.005$ | $0.027 \pm 0.001$ | $0.106 \pm 0.003$ | $0.579 \pm 0.003$ | $0.096 \pm 0.003$ | $0.118 \pm 0.002$ | $0.329 \pm 0.005$ | $0.421 \pm 0.005$ |
| | G2P2 | $0.453 \pm 0.002$ | $0.201 \pm 0.001$ | $0.453 \pm 0.001$ | $0.644 \pm 0.004$ | $\underline{0.322 \pm 0.003}$ | $\mathbf{0.452 \pm 0.001}$ | $0.576 \pm 0.006$ | $\underline{0.436 \pm 0.001}$ |
| | TAGA | $\mathbf{0.537 \pm 0.003}$ | $\mathbf{0.224 \pm 0.001}$ | $\mathbf{0.498 \pm 0.004}$ | $\mathbf{0.682 \pm 0.005}$ | $\mathbf{0.351 \pm 0.009}$ | $\underline{0.419 \pm 0.001}$ | $\mathbf{0.616 \pm 0.009}$ | $\mathbf{0.448 \pm 0.003}$ |
| | TAGA-rw | $\underline{0.530 \pm 0.001}$ | $\underline{0.221 \pm 0.001}$ | $\underline{0.494 \pm 0.001}$ | $\underline{0.680 \pm 0.002}$ | $0.301 \pm 0.003$ | $0.394 \pm 0.001$ | $\underline{0.599 \pm 0.002}$ | $0.434 \pm 0.002$ |
| 1 | PLM | $0.280 \pm 0.044$ | $0.122 \pm 0.042$ | $0.238 \pm 0.039$ | $0.412 \pm 0.080$ | $0.284 \pm 0.078$ | $0.230 \pm 0.051$ | $0.503 \pm 0.067$ | $0.282 \pm 0.068$ |
| | GraphMAE | $0.255 \pm 0.041$ | $0.128 \pm 0.028$ | $0.300 \pm 0.052$ | $0.474 \pm 0.058$ | $0.231 \pm 0.052$ | $0.304 \pm 0.066$ | $0.492 \pm 0.076$ | $0.270 \pm 0.042$ |
| | GraphCL | $0.123 \pm 0.031$ | $0.157 \pm 0.066$ | $0.256 \pm 0.039$ | $0.402 \pm 0.059$ | $0.371 \pm 0.124$ | $0.325 \pm 0.079$ | $0.414 \pm 0.040$ | $0.347 \pm 0.079$ |
| | GRACE | $0.263 \pm 0.034$ | $0.138 \pm 0.035$ | $0.336 \pm 0.051$ | $0.435 \pm 0.071$ | $0.266 \pm 0.085$ | $0.295 \pm 0.053$ | $0.514 \pm 0.095$ | $0.282 \pm 0.045$ |
| | GraphFormers | $0.233 \pm 0.042$ | $0.131 \pm 0.038$ | $0.247 \pm 0.052$ | $0.463 \pm 0.069$ | $0.231 \pm 0.055$ | $0.284 \pm 0.043$ | $0.471 \pm 0.054$ | $0.284 \pm 0.057$ |
| | PATTON | $0.217 \pm 0.059$ | $0.127 \pm 0.042$ | $0.305 \pm 0.048$ | $0.487 \pm 0.057$ | $0.286 \pm 0.078$ | $0.318 \pm 0.053$ | $0.523 \pm 0.051$ | $0.243 \pm 0.068$ |
| | G2P2 | $0.308 \pm 0.052$ | $0.145 \pm 0.029$ | $0.359 \pm 0.044$ | $0.477 \pm 0.082$ | $0.361 \pm 0.092$ | $0.372 \pm 0.066$ | $0.522 \pm 0.085$ | $0.356 \pm 0.042$ |
| | TAGA | $\mathbf{0.323 \pm 0.040}$ | $\mathbf{0.180 \pm 0.073}$ | $\mathbf{0.380 \pm 0.062}$ | $\underline{0.509 \pm 0.089}$ | $\mathbf{0.413 \pm 0.114}$ | $\mathbf{0.417 \pm 0.077}$ | $\mathbf{0.563 \pm 0.062}$ | $\underline{0.440 \pm 0.070}$ |
| | TAGA-rw | $\underline{0.307 \pm 0.050}$ | $\underline{0.171 \pm 0.013}$ | $\underline{0.365 \pm 0.042}$ | $\mathbf{0.561 \pm 0.063}$ | $\underline{0.383 \pm 0.078}$ | $\underline{0.380 \pm 0.037}$ | $\underline{0.548 \pm 0.073}$ | $\mathbf{0.498 \pm 0.084}$ |
| 5 | PLM | $0.500 \pm 0.019$ | $0.210 \pm 0.025$ | $0.377 \pm 0.027$ | $0.641 \pm 0.031$ | $0.557 \pm 0.040$ | $0.420 \pm 0.037$ | $0.632 \pm 0.040$ | $0.478 \pm 0.056$ |
| | GraphMAE | $0.425 \pm 0.028$ | $0.212 \pm 0.029$ | $0.434 \pm 0.036$ | $0.704 \pm 0.038$ | $0.459 \pm 0.038$ | $0.489 \pm 0.038$ | $0.625 \pm 0.049$ | $0.452 \pm 0.037$ |
| | GraphCL | $0.231 \pm 0.015$ | $0.201 \pm 0.040$ | $0.397 \pm 0.040$ | $0.641 \pm 0.044$ | $0.531 \pm 0.047$ | $0.462 \pm 0.041$ | $0.584 \pm 0.037$ | $0.477 \pm 0.048$ |
| | GRACE | $0.445 \pm 0.028$ | $0.227 \pm 0.031$ | $0.472 \pm 0.040$ | $0.685 \pm 0.027$ | $0.481 \pm 0.061$ | $0.515 \pm 0.042$ | $0.628 \pm 0.047$ | $0.482 \pm 0.040$ |
| | GraphFormers | $0.461 \pm 0.022$ | $0.230 \pm 0.031$ | $0.374 \pm 0.031$ | $0.731 \pm 0.029$ | $0.458 \pm 0.045$ | $0.498 \pm 0.032$ | $0.619 \pm 0.039$ | $0.568 \pm 0.053$ |
| | PATTON | $0.471 \pm 0.039$ | $0.227 \pm 0.040$ | $0.405 \pm 0.032$ | $0.699 \pm 0.025$ | $0.466 \pm 0.038$ | $0.518 \pm 0.030$ | $0.605 \pm 0.042$ | $0.532 \pm 0.048$ |
| | G2P2 | $0.466 \pm 0.025$ | $0.240 \pm 0.034$ | $\underline{0.510 \pm 0.039}$ | $0.703 \pm 0.032$ | $0.617 \pm 0.053$ | $0.583 \pm 0.051$ | $0.640 \pm 0.051$ | $0.565 \pm 0.055$ |
| | TAGA | $\mathbf{0.483 \pm 0.022}$ | $\underline{0.263 \pm 0.031}$ | $\mathbf{0.543 \pm 0.038}$ | $\underline{0.752 \pm 0.028}$ | $\mathbf{0.636 \pm 0.046}$ | $\mathbf{0.602 \pm 0.041}$ | $\underline{0.649 \pm 0.044}$ | $\underline{0.664 \pm 0.061}$ |
| | TAGA-rw | $\underline{0.471 \pm 0.031}$ | $\mathbf{0.276 \pm 0.053}$ | $0.508 \pm 0.019$ | $\mathbf{0.764 \pm 0.027}$ | $\underline{0.621 \pm 0.076}$ | $\underline{0.594 \pm 0.025}$ | $\mathbf{0.684 \pm 0.027}$ | $\mathbf{0.675 \pm 0.070}$ |
| 10 | PLM | $\underline{0.526 \pm 0.013}$ | $0.240 \pm 0.018$ | $0.463 \pm 0.029$ | $0.690 \pm 0.017$ | $0.639 \pm 0.038$ | $0.491 \pm 0.028$ | $0.679 \pm 0.023$ | $0.535 \pm 0.038$ |
| | GraphMAE | $0.461 \pm 0.017$ | $0.234 \pm 0.014$ | $0.511 \pm 0.028$ | $0.761 \pm 0.023$ | $0.535 \pm 0.042$ | $0.543 \pm 0.035$ | $0.659 \pm 0.028$ | $0.508 \pm 0.028$ |
| | GraphCL | $0.301 \pm 0.018$ | $0.233 \pm 0.029$ | $0.488 \pm 0.031$ | $0.702 \pm 0.025$ | $0.566 \pm 0.043$ | $0.523 \pm 0.044$ | $0.632 \pm 0.025$ | $0.531 \pm 0.035$ |
| | GRACE | $0.488 \pm 0.018$ | $0.251 \pm 0.015$ | $0.552 \pm 0.028$ | $0.754 \pm 0.018$ | $0.567 \pm 0.054$ | $0.567 \pm 0.031$ | $0.670 \pm 0.025$ | $0.529 \pm 0.033$ |
| | GraphFormers | $0.482 \pm 0.019$ | $0.248 \pm 0.030$ | $0.447 \pm 0.028$ | $0.778 \pm 0.022$ | $0.498 \pm 0.035$ | $0.538 \pm 0.026$ | $0.633 \pm 0.034$ | $0.601 \pm 0.040$ |
| | PATTON | $0.501 \pm 0.028$ | $0.247 \pm 0.024$ | $0.451 \pm 0.026$ | $0.738 \pm 0.020$ | $0.533 \pm 0.029$ | $0.539 \pm 0.028$ | $0.643 \pm 0.028$ | $0.564 \pm 0.041$ |
| | G2P2 | $\mathbf{0.527 \pm 0.014}$ | $0.269 \pm 0.018$ | $\underline{0.598 \pm 0.031}$ | $0.753 \pm 0.020$ | $0.649 \pm 0.046$ | $\underline{0.632 \pm 0.037}$ | $\underline{0.691 \pm 0.029}$ | $0.618 \pm 0.037$ |
| | TAGA | $0.521 \pm 0.017$ | $\underline{0.288 \pm 0.025}$ | $\mathbf{0.622 \pm 0.025}$ | $0.788 \pm 0.021$ | $\mathbf{0.679 \pm 0.041}$ | $\mathbf{0.651 \pm 0.048}$ | $\mathbf{0.714 \pm 0.024}$ | $\mathbf{0.705 \pm 0.045}$ |
| | TAGA-rw | $0.518 \pm 0.010$ | $\mathbf{0.288 \pm 0.040}$ | $0.595 \pm 0.024$ | $\mathbf{0.806 \pm 0.011}$ | $\underline{0.652 \pm 0.046}$ | $0.626 \pm 0.020$ | $0.679 \pm 0.013$ | $0.662 \pm 0.056$ |

Table 1: Performance in zero-shot and few-shot node classification for each dataset and setting. The best-performing model is highlighted in **bold**, and the second-best performing model is underlined.

embeddings for robust results. Commonly used GNN models (GCN (Kipf & Welling, 2017), GIN (Hamilton et al., 2017), GraphSAGE (Xu et al., 2018)) are chosen as the backbone model as the backbone model for both our method and all comparison methods. For a fair comparison, all models are required to adhere to the same GNN architecture, including the number of convolution layers and hidden dimensions. More details about hyperparameters can be found in Appendix B.1. Further technical details can be found in Appendix C. Our code can be found at anonymous link `https://anonymous.4open.science/r/TAGA-32B7/`.

## 5.2 EFFECTIVENESS RESULTS

In this section, we assess the effectiveness of our proposed unsupervised representation learning framework compared to other methods under conditions of label scarcity. Our representation learning models are initially pre-trained on each TAG dataset without any supervised labels. After the pre-training phase, we evaluate the quality of the obtained node embeddings under zero-shot conditions by measuring the similarity between these embeddings and the corresponding text label embeddings. To further gauge performance in scenarios with limited labeled data, we conduct evaluations using 1, 3, 5, 10, 20, 50, and 100-shot settings. Due to space limitation, the results with text encoder `UAE-Large-V1` under zero-shot and 1, 5, 10-shot settings is reported in Table 1. Our acceleration method with random walk is denoted as "TAGA-rw". The results with `text-embedding-3-small` and other few-shot settings can be found in Appendix B.4. We also present zero-shot link prediction performance in Appendix B.3.

**Zero-shot performance.** Table 1 presents node classification accuracy under zero-shot conditions, where our method consistently outperforms all comparison methods in seven out of eight datasets. On average, our

method surpasses other graph pre-training methods by 47.84% and exceeds the second-best model by 6.78%. These findings demonstrate the enhanced ability of our pre-trained model to effectively learn representations that enable zero-shot predictions. Furthermore, compared to direct textual embeddings from the PLM, our method improves zero-shot performance by an average of 20.76%. This demonstrates our method's capacity in integrating structural and textual information from neighborhoods over directly using the PLM. Interestingly, our method exhibits a stronger performance advantage when dealing with data rich in textual information. Specifically, for the two citation networks (Arxiv and Cora), which possess significantly longer text attributes compared to other datasets, our method surpasses the second-best performing graph pretrained model by an average of 10.33%. This proves our method can effectively leverage the rich textual information.

**Few-shot performance.** For few-shot experiments, our method consistently outperforms all comparison methods, achieving a 15.55% average improvement and surpassing the second-best model by 6.28% on average. Notably, our method exhibits a more pronounced advantage in scenarios with limited labeled data (¡=5 shots), where it outperforms all other methods by an average of 19.79% and exceeds the second-best model by 7.91% on average. This underscores the effectiveness of our method, particularly in settings where few-shot learning is essential due to data labels constraints.

**Remarks.** It is worth noting that for some datasets, the zero-shot performance of our method can match or even exceed few-shot predictive results, particularly when the number of training samples for few-shot learning is limited. For example, on five datasets (Arxiv, Children, Computers, Cora, and Pubmed), the zero-shot performance surpasses 1-shot performance by an average of 23.54%. Remarkably, the zero-shot performance can even be comparable to that of 5-shot. This demonstrates the strong potential of our method in scenarios where labeled data is scarce or unreachable.

### 5.3 TRANSFER ABILITY ANALYSIS

In real-world applications, not only labels are difficult to obtain, but the data itself is also scarce. This necessitates the generalization of a pre-trained model to a data domain distinct from the pre-training data. Here we evaluate the zero-shot and few-shot performance under transfer learning settings. Specifically, the model is unsupervisedly pre-trained on the source data domain and then transferred to the target data domain. No further fine-tuning is performed for zero-shot prediction, and is fine-tuned using the limited training samples for few-shot prediction.

| | Source | Cora ↓ Arxiv | Arxiv ↓ Cora | Cora ↓ Pubmed | Pubmed ↓ Cora | Children ↓ History | History ↓ Children | Computers ↓ Photo | Photo ↓ Computers |
|---|---|---|---|---|---|---|---|---|---|
| | Target | | | | | | | | |
| 0-shot | GRACE | 0.021 | 0.173 | 0.360 | 0.302 | 0.073 | 0.065 | 0.099 | 0.070 |
| | GraphMAE | 0.012 | 0.153 | 0.434 | 0.239 | 0.009 | 0.030 | 0.082 | 0.004 |
| | GraphCL | 0.015 | 0.232 | 0.368 | 0.178 | 0.045 | 0.024 | 0.094 | 0.135 |
| | G2P2 | 0.241 | 0.647 | 0.421 | 0.533 | **0.204** | 0.100 | 0.297 | 0.340 |
| | TAGA | **0.406** | **0.679** | **0.484** | **0.559** | 0.184 | 0.200 | 0.452 | **0.372** |
| | TAGA-rw | 0.398 | 0.624 | 0.408 | 0.526 | 0.176 | **0.203** | **0.455** | 0.348 |
| 5-shot | GRACE | 0.426 | 0.721 | 0.591 | 0.657 | 0.609 | 0.219 | 0.483 | 0.382 |
| | GraphMAE | 0.426 | 0.645 | 0.578 | 0.515 | 0.527 | 0.160 | 0.367 | 0.294 |
| | GraphCL | 0.107 | 0.678 | 0.436 | 0.416 | 0.598 | 0.178 | 0.395 | 0.345 |
| | G2P2 | 0.395 | 0.749 | 0.633 | 0.708 | 0.623 | 0.239 | 0.509 | 0.429 |
| | TAGA | **0.475** | 0.754 | **0.655** | **0.734** | **0.651** | **0.257** | **0.528** | **0.448** |
| | TAGA-rw | 0.443 | **0.764** | 0.644 | 0.674 | 0.617 | 0.250 | 0.482 | 0.436 |

Table 2: Transfer learning results. The best-performing model is highlighted in **bold**.

In Table 2, we present the performance of zero-shot and five-shot predictions across eight pairs of source and target datasets. The results demonstrate a clear advantage for our method in the zero-shot setting, where it consistently outperforms all other methods across all dataset pairs. Notably, our method achieves an average improvement of 26.5% over the second-best performing method. In the five-shot setting, our method continues outperforming the second-best performing method by 4.53% on average. Particularly when transferring from Cora to Arxiv and Pubmed, and Children to History, our method achieves significant performance gain by 6.30% on average, demonstrating its ability to effectively leverage limited labeled data in the target domain.

### 5.4 ABLATION STUDY

To investigate the effectiveness of our proposed model compared to simpler heuristics, we conducted a series of ablation analyses. We began by considering textual embeddings obtained directly by applying the PLM to the Text of Graph views' corpus at different orders. This allowed us to assess the impact of our

training procedure compared to a simpler approach that relies solely on Text-of-Graph view representations. In addition, we compare our full model with a variant, *Glo-GofT*, which only aligns the GNN embeddings that aggregate individual node's text embeddings but removes all higher-order Graph-of-Text embeddings. The results of these ablation studies are presented in Table 3, which reveals that removing compo-

|  | Method | arxiv | children | computers | cora | history | photo | pubmed | sports |
|---|---|---|---|---|---|---|---|---|---|
| 0-shot | Full | **0.537** | **0.224** | 0.498 | **0.682** | **0.351** | **0.419** | **0.616** | **0.448** |
|  | TofG-0 | 0.500 | 0.099 | 0.423 | 0.575 | 0.318 | 0.392 | 0.471 | 0.318 |
|  | TofG-1 | 0.521 | 0.102 | 0.544 | 0.601 | 0.349 | 0.336 | 0.512 | 0.444 |
|  | TofG-2 | 0.519 | 0.098 | **0.556** | 0.606 | 0.348 | 0.327 | 0.532 | **0.448** |
|  | Glo-GofT | 0.533 | 0.205 | 0.482 | 0.657 | 0.329 | 0.407 | 0.522 | 0.417 |
| 5-shot | Full | 0.483 | **0.263** | 0.543 | **0.752** | **0.636** | **0.602** | **0.649** | **0.664** |
|  | TofG-0 | **0.500** | 0.210 | 0.377 | 0.641 | 0.557 | 0.420 | 0.632 | 0.478 |
|  | TofG-1 | 0.496 | 0.234 | 0.549 | 0.709 | 0.598 | 0.582 | 0.631 | 0.615 |
|  | TofG-2 | 0.490 | 0.234 | **0.558** | 0.706 | 0.589 | 0.590 | 0.631 | 0.654 |
|  | Glo-GofT | 0.479 | 0.257 | 0.512 | 0.726 | 0.623 | 0.592 | 0.635 | 0.629 |

Figure 3: Ablation studies results of zero- and five-shot settings. Here "Full" denotes our full model.

nents of our full model generally leads to a decrease in performance. In the zero-shot setting, the full model outperforms the variant models by 2.79% to 8.49% on average, and ranges from 1.74% to 9.71% in the five-shot setting. These results underscore the contribution of each component to TAGA's overall effectiveness. In Appendix B.5, we have shown additional ablation studies that evaluate how will aligning on different orders of hierarchies will influence the representation due to space limitation.

## 5.5 EFFICIENCY ANALYSIS

To validate the efficiency and scalability of our proposed full method and random walk algorithm

during both training and inference phases, we conduct experiments on the Cora dataset. We vary the number of hops from 0 to 5 and record the number of words in the input corpus, training time, and inference time. The results are presented in Figure 4. As depicted in top figure, the exponential growth in input size for the full method compared to the near-linear growth of the random walk method demonstrates the our's superior scalability in managing larger graph neighborhoods. The middle figure further demonstrates the efficiency advantage of the random walk algorithm, as its training time increases linearly with the number of hops, whereas the full method experiences a much steeper increase, becoming infeasible beyond 3 hops due to out-of-memory (OOM) errors. Finally, the bottom figure highlights the speedup achieved by our proposed method during inference compared to directly using a PLM. The inference time for our method remains linear growth trend across different hops, while the PLM-based approach suffers from rapidly increasing inference time with the hops number.

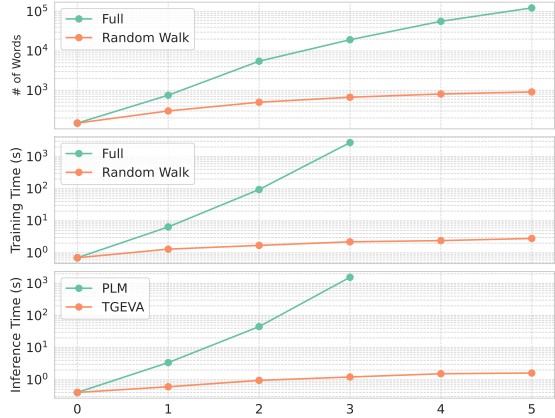

Figure 4: (top) Comparison of the full method and the random walk algorithm in terms of the number of words, and (middle) training time, and (bottom) inference time comparison between PLM and TAGA in terms of the number of hops.

## 6 CONCLUSIONS

In this paper, we introduce TAGA, a novel self-supervised learning framework designed to address the challenges of unsupervised representation learning on TAGs. TAGA integrates both textual and structural information within TAGs by aligning representations from two complementary views: *Text-of-Graph* and *Graph-of-Text*. To enhance the preservation of structural information in the *Text-of-Graph* view, we propose a natural hierarchical document layout that mirrors the graph's topology. Additionally, we introduce a structure-preserving random walk algorithm to accelerate the training process on large TAGs. Extensive experiments on eight real-world datasets demonstrate TAGA's superior performance in zero-shot and few-shot learning scenarios, showcasing its strong generalization capabilities across diverse domains.

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

## A    ADDITIONAL RELATED WORKS

### A.1    EFFICIENT AND SCALABLE METHODS FOR LARGE-SIZE GRAPH NEIGHBORHOODS

Efficiency and scalability are crucial for deep graph learning, particularly when dealing with large graphs or high-order interactions. Traditional graph sampling techniques, such as node sampling (Chen et al., 2018), edge sampling (Hamilton et al., 2017), or subgraph sampling (Zeng et al., 2019), aim to reduce neighborhood size. However, these methods may not be suitable for TAGs, as they can result in the loss of important hierarchical interactive connection during the random sampling process. Meanwhile, in the NLP domain, some efforts (Peng et al., 2023; Han et al., 2023; Chen et al., 2023a; Jiang et al., 2023; Ding et al., 2023a) have been made to address the long context issue of PLMs. These approaches typically involve compressing input tokens into latent vectors (Jiang et al., 2023) or modifying the attention mask (Chen et al., 2023b; Han et al., 2023; Ding et al., 2023a) to reduce significant interactions. However, these methods often fail to preserve the original structure of the input corpus and might alter the hierarchical layout.

## B    ADDITIONAL EXPERIMENTAL RESULTS AND SETTINGS

In this section, we present additional experimental settings and results due to the space limitation of the main paper.

### B.1    ADDITIONAL IMPLEMENTATION SETTINGS

All experiments are conducted on a 64-bit machine with four 16GB NVIDIA GPUs. Each experiment involves running the models 20 times with different random seeds to minimize variance due to specific data splits. Accuracy is adopted as the evaluation metric for node classification tasks. Specifically, for smaller datasets such as Cora and PubMed, we employ 3 convolution layers, while for larger datasets, we utilize 2 layers. Latent dimension is aligned with the PLM embedding dimension. During the pre-train stage, the model is trained with 40,000 steps on each dataset with minibatch size 8. The learning rate is initialized as $1e^{-3}$ and with decay rate 0.999 each 10 steps. For zero-shot predictions, we utilize the entire dataset as the test set. In the case of $k$-shot predictions, we randomly select $k$ samples from each class to form the training set, dividing the remaining data into validation and test sets at a ratio of 1:9. All models undergo finetune for 100 epochs, and testing is based on the best validation results.

### B.2    SENSITIVITY ANALYSIS

In this section, we investigate the sensitivity of the key hyperparameters and their impact on TAGA's performance. Specifically, we first evaluate how different GNN backbones (GCN, GIN, and GraphSAGE) affect performance. Then we evaluate how jumping ratio ($p$) and maximum walk length ($L$) would affect random walk's performance. The results are presented in Figure 5. The sensitivity analysis conducted on TAGA's performance demonstrates that the method is robust across a range of hyperparameters. Specifically, the variance in performance across different GNN backbones is 0.84%, indicating a stable behavior regardless of the backbone employed. Similarly, adjustments in the jumping ratio ($p$) and maximum walk length ($L$) exhibit 0.33% and 0.76% variance on average, which underscores that our method is not sensitive to the hyperparameters chosen.

### B.3    ADDITIONAL LINK PREDICTION EXPERIMENTS

In order to verify the generalizability of our method, the transfer learning setting is adopted. The representation learning method is pre-trained on source dataset, and then directly perform link prediction task on target dataset without any finetune process. The ratio of positive and negative edges is 1:1 and we use cosine similarity to measure the scores. From the Table 3 we can observe that our proposed method outperforms all the comparison methods in 15 out of 16 tasks on ROC-AUC metric, which further verified the effectiveness and generalizability of our proposed representation learning method.

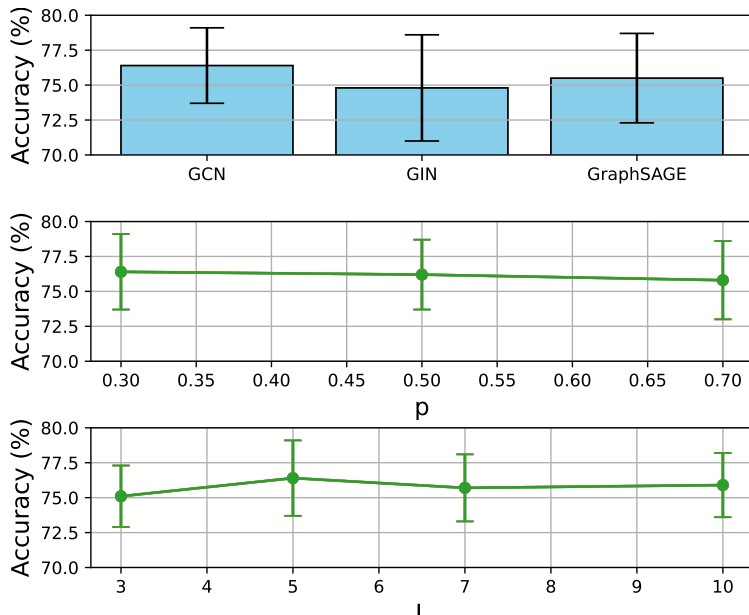

Figure 5: Comparison of five-shot performance between (top) different GNN encoder choices, and (middle) varying jumping ratio, and (bottom) maximum walk length of random walks.

| Source | Target | GRACE | G2P2 | TAGA |
|---|---|---|---|---|
| Pubmed | Cora | $0.6007 \pm 0.0019$ | $0.9964 \pm 0.0001$ | $\mathbf{0.9971 \pm 0.0005}$ |
| | Pubmed | $0.8240 \pm 0.0008$ | $0.9564 \pm 0.0003$ | $\mathbf{0.9683 \pm 0.0002}$ |
| | Sports | $0.6094 \pm 0.0002$ | $\mathbf{0.9864 \pm 0.0000}$ | $0.9844 \pm 0.0000$ |
| | Arxiv | $0.5318 \pm 0.0002$ | $0.9847 \pm 0.0000$ | $\mathbf{0.9865 \pm 0.0001}$ |
| Arxiv | Cora | $0.9170 \pm 0.0008$ | $0.9928 \pm 0.0002$ | $\mathbf{0.9947 \pm 0.0003}$ |
| | Pubmed | $0.8047 \pm 0.0006$ | $0.9563 \pm 0.0003$ | $\mathbf{0.9662 \pm 0.0004}$ |
| | Sports | $0.7636 \pm 0.0001$ | $0.9907 \pm 0.0000$ | $\mathbf{0.9940 \pm 0.0000}$ |
| | Arxiv | $0.9386 \pm 0.0001$ | $0.9857 \pm 0.0000$ | $\mathbf{0.9886 \pm 0.0000}$ |
| Cora | Cora | $0.9646 \pm 0.0005$ | $0.9886 \pm 0.0004$ | $\mathbf{0.9959 \pm 0.0002}$ |
| | Pubmed | $0.9363 \pm 0.0006$ | $0.9508 \pm 0.0005$ | $\mathbf{0.9634 \pm 0.0002}$ |
| | Sports | $0.9727 \pm 0.0000$ | $0.9816 \pm 0.0000$ | $\mathbf{0.9913 \pm 0.0000}$ |
| | Arxiv | $0.9735 \pm 0.0001$ | $0.9620 \pm 0.0001$ | $\mathbf{0.9901 \pm 0.0000}$ |
| Sports | Cora | $0.7847 \pm 0.0010$ | $0.9911 \pm 0.0002$ | $\mathbf{0.9955 \pm 0.0002}$ |
| | Pubmed | $0.8718 \pm 0.0005$ | $0.9611 \pm 0.0003$ | $\mathbf{0.9667 \pm 0.0005}$ |
| | Sports | $0.9353 \pm 0.0000$ | $0.9906 \pm 0.0000$ | $\mathbf{0.9942 \pm 0.0000}$ |
| | Arxiv | $0.8990 \pm 0.0001$ | $0.9780 \pm 0.0000$ | $\mathbf{0.9842 \pm 0.0000}$ |

Table 3: The ROC-AUC experimental results of zero-shot link prediction tasks by transferring from the source dataset to target dataset.

### B.4 ADDITIONAL NODE CLASSIFICATION ANALYSIS

We present additional zero-shot and few-shot performance under two different text encoders `UAE-Large-V1` and `Text-embedding-3-small`. The zero-shot results are present in Table 5. The few-shot results with text encoder `UAE-Large-V1` is present in Table 6, and few-shot results with text encoder `Text-embedding-3-small` is present in Table 7. From the table, we can observe that our method TAGA consistently achieve the best performance on two different choices of text encoder models. This demonstrates the effectiveness and robustness of our proposed method.

| k-Shot | Model | Arxiv | Children | Computers | Cora | History | Photo | Pubmed | Sports |
|---|---|---|---|---|---|---|---|---|---|
|  | # Nodes | 169,343 | 76,875 | 87,229 | 2,708 | 41,551 | 48,362 | 19,717 | 173,055 |
|  | # Edges | 1,166,243 | 1,554,578 | 721,107 | 10,556 | 358,574 | 500,939 | 44,338 | 1,773,594 |
|  | Avg # Words | 220.7 | 199.3 | 90.7 | 148.2 | 218.7 | 144.5 | 50.1 | 9.8 |
| 0 | PLM | 0.500 ± 0.001 | 0.094 ± 0.003 | 0.427 ± 0.001 | 0.624 ± 0.005 | 0.169 ± 0.001 | 0.387 ± 0.009 | 0.475 ± 0.008 | 0.316 ± 0.002 |
|  | GraphMAE | 0.104 ± 0.001 | 0.021 ± 0.001 | 0.049 ± 0.001 | 0.194 ± 0.006 | 0.019 ± 0.001 | 0.152 ± 0.001 | 0.438 ± 0.001 | 0.112 ± 0.001 |
|  | GraphCL | 0.089 ± 0.001 | 0.037 ± 0.001 | 0.173 ± 0.001 | 0.176 ± 0.003 | 0.191 ± 0.001 | 0.174 ± 0.001 | 0.368 ± 0.001 | 0.140 ± 0.001 |
|  | GRACE | 0.045 ± 0.001 | 0.034 ± 0.001 | 0.169 ± 0.001 | 0.146 ± 0.004 | 0.079 ± 0.001 | 0.025 ± 0.001 | 0.335 ± 0.001 | 0.057 ± 0.001 |
|  | G2P2 | 0.453 ± 0.002 | 0.201 ± 0.001 | 0.453 ± 0.001 | 0.644 ± 0.004 | 0.322 ± 0.003 | **0.452 ± 0.001** | 0.576 ± 0.006 | 0.436 ± 0.001 |
|  | TAGA | **0.537 ± 0.003** | **0.224 ± 0.001** | **0.498 ± 0.004** | **0.682 ± 0.005** | **0.351 ± 0.009** | 0.419 ± 0.001 | **0.616 ± 0.009** | **0.448 ± 0.003** |
|  | TAGA-rw | 0.530 ± 0.001 | 0.221 ± 0.001 | 0.494 ± 0.001 | 0.680 ± 0.002 | 0.301 ± 0.003 | 0.394 ± 0.001 | 0.599 ± 0.002 | 0.434 ± 0.002 |
| 1 | PLM | 0.280 ± 0.044 | 0.122 ± 0.042 | 0.238 ± 0.039 | 0.412 ± 0.080 | 0.284 ± 0.078 | 0.230 ± 0.051 | 0.503 ± 0.067 | 0.282 ± 0.068 |
|  | GraphMAE | 0.255 ± 0.041 | 0.128 ± 0.028 | 0.300 ± 0.052 | 0.474 ± 0.058 | 0.231 ± 0.052 | 0.304 ± 0.066 | 0.492 ± 0.076 | 0.270 ± 0.042 |
|  | GraphCL | 0.123 ± 0.031 | 0.157 ± 0.066 | 0.256 ± 0.039 | 0.402 ± 0.059 | 0.371 ± 0.124 | 0.325 ± 0.079 | 0.414 ± 0.040 | 0.347 ± 0.079 |
|  | GRACE | 0.263 ± 0.034 | 0.138 ± 0.035 | 0.336 ± 0.051 | 0.435 ± 0.071 | 0.266 ± 0.085 | 0.295 ± 0.053 | 0.514 ± 0.095 | 0.282 ± 0.045 |
|  | G2P2 | 0.308 ± 0.052 | 0.145 ± 0.029 | 0.359 ± 0.044 | 0.477 ± 0.082 | 0.361 ± 0.092 | 0.372 ± 0.066 | 0.522 ± 0.085 | 0.356 ± 0.042 |
|  | TAGA | **0.323 ± 0.040** | **0.180 ± 0.073** | **0.380 ± 0.062** | 0.509 ± 0.089 | **0.413 ± 0.114** | **0.417 ± 0.077** | **0.563 ± 0.062** | 0.440 ± 0.070 |
|  | TAGA-rw | 0.307 ± 0.050 | 0.171 ± 0.013 | 0.365 ± 0.042 | **0.561 ± 0.063** | 0.383 ± 0.078 | 0.380 ± 0.037 | 0.548 ± 0.073 | **0.498 ± 0.084** |
| 3 | PLM | 0.436 ± 0.036 | 0.194 ± 0.029 | 0.318 ± 0.038 | 0.588 ± 0.036 | 0.448 ± 0.071 | 0.352 ± 0.044 | 0.611 ± 0.051 | 0.392 ± 0.041 |
|  | GraphMAE | 0.379 ± 0.039 | 0.182 ± 0.025 | 0.389 ± 0.035 | 0.634 ± 0.044 | 0.362 ± 0.050 | 0.432 ± 0.051 | 0.597 ± 0.061 | 0.363 ± 0.050 |
|  | GraphCL | 0.192 ± 0.029 | 0.186 ± 0.039 | 0.343 ± 0.046 | 0.563 ± 0.044 | 0.484 ± 0.071 | 0.382 ± 0.052 | 0.476 ± 0.038 | 0.373 ± 0.071 |
|  | GRACE | 0.398 ± 0.031 | 0.200 ± 0.038 | 0.442 ± 0.045 | 0.622 ± 0.043 | 0.404 ± 0.057 | 0.447 ± 0.053 | 0.620 ± 0.055 | 0.398 ± 0.045 |
|  | G2P2 | 0.430 ± 0.027 | 0.207 ± 0.038 | 0.469 ± 0.042 | 0.623 ± 0.033 | 0.508 ± 0.073 | 0.528 ± 0.049 | 0.641 ± 0.064 | 0.464 ± 0.050 |
|  | TAGA | **0.445 ± 0.035** | **0.241 ± 0.062** | **0.497 ± 0.035** | 0.695 ± 0.050 | 0.551 ± 0.094 | **0.551 ± 0.045** | 0.659 ± 0.058 | **0.586 ± 0.057** |
|  | TAGA-rw | 0.442 ± 0.040 | 0.222 ± 0.060 | 0.467 ± 0.025 | **0.705 ± 0.021** | 0.558 ± 0.072 | 0.513 ± 0.070 | **0.632 ± 0.043** | 0.569 ± 0.105 |
| 5 | PLM | 0.500 ± 0.019 | 0.210 ± 0.025 | 0.377 ± 0.027 | 0.641 ± 0.031 | 0.557 ± 0.040 | 0.420 ± 0.037 | 0.632 ± 0.040 | 0.478 ± 0.056 |
|  | GraphMAE | 0.425 ± 0.028 | 0.212 ± 0.029 | 0.434 ± 0.036 | 0.704 ± 0.038 | 0.459 ± 0.038 | 0.489 ± 0.038 | 0.625 ± 0.049 | 0.452 ± 0.037 |
|  | GraphCL | 0.231 ± 0.015 | 0.201 ± 0.040 | 0.397 ± 0.040 | 0.641 ± 0.044 | 0.531 ± 0.047 | 0.462 ± 0.041 | 0.584 ± 0.037 | 0.477 ± 0.048 |
|  | GRACE | 0.445 ± 0.028 | 0.227 ± 0.031 | 0.472 ± 0.040 | 0.685 ± 0.027 | 0.481 ± 0.061 | 0.515 ± 0.042 | 0.628 ± 0.047 | 0.482 ± 0.040 |
|  | G2P2 | 0.466 ± 0.025 | 0.240 ± 0.034 | 0.510 ± 0.039 | 0.703 ± 0.032 | 0.617 ± 0.053 | 0.583 ± 0.051 | 0.640 ± 0.051 | 0.565 ± 0.055 |
|  | TAGA | **0.483 ± 0.022** | 0.263 ± 0.031 | **0.543 ± 0.038** | 0.752 ± 0.028 | **0.636 ± 0.046** | 0.602 ± 0.041 | 0.649 ± 0.044 | 0.664 ± 0.061 |
|  | TAGA-rw | 0.471 ± 0.031 | **0.276 ± 0.053** | 0.508 ± 0.019 | **0.764 ± 0.027** | 0.621 ± 0.076 | **0.594 ± 0.025** | **0.684 ± 0.027** | **0.675 ± 0.070** |
| 10 | PLM | 0.526 ± 0.013 | 0.240 ± 0.018 | 0.463 ± 0.029 | 0.690 ± 0.017 | 0.639 ± 0.038 | 0.491 ± 0.028 | 0.679 ± 0.023 | 0.535 ± 0.038 |
|  | GraphMAE | 0.461 ± 0.017 | 0.234 ± 0.014 | 0.511 ± 0.028 | 0.761 ± 0.023 | 0.535 ± 0.042 | 0.543 ± 0.035 | 0.659 ± 0.028 | 0.508 ± 0.028 |
|  | GraphCL | 0.301 ± 0.018 | 0.233 ± 0.029 | 0.488 ± 0.031 | 0.702 ± 0.025 | 0.566 ± 0.043 | 0.523 ± 0.044 | 0.632 ± 0.025 | 0.531 ± 0.035 |
|  | GRACE | 0.488 ± 0.018 | 0.251 ± 0.015 | 0.552 ± 0.028 | 0.754 ± 0.018 | 0.567 ± 0.054 | 0.567 ± 0.031 | 0.670 ± 0.025 | 0.529 ± 0.033 |
|  | G2P2 | **0.527 ± 0.014** | 0.269 ± 0.018 | 0.598 ± 0.031 | 0.753 ± 0.020 | 0.649 ± 0.046 | 0.632 ± 0.037 | 0.691 ± 0.029 | 0.618 ± 0.037 |
|  | TAGA | 0.521 ± 0.017 | 0.288 ± 0.025 | **0.622 ± 0.025** | 0.788 ± 0.021 | **0.679 ± 0.041** | **0.651 ± 0.048** | **0.714 ± 0.024** | **0.705 ± 0.045** |
|  | TAGA-rw | 0.518 ± 0.010 | 0.288 ± 0.040 | 0.595 ± 0.024 | **0.806 ± 0.011** | 0.652 ± 0.046 | 0.626 ± 0.020 | 0.679 ± 0.013 | 0.662 ± 0.056 |
| 100 | PLM | 0.592 ± 0.005 | 0.337 ± 0.013 | 0.610 ± 0.008 | 0.753 ± 0.014 | 0.753 ± 0.008 | 0.634 ± 0.015 | 0.771 ± 0.005 | 0.690 ± 0.013 |
|  | GraphMAE | 0.573 ± 0.005 | 0.319 ± 0.008 | 0.650 ± 0.008 | 0.835 ± 0.007 | 0.684 ± 0.011 | 0.655 ± 0.012 | 0.744 ± 0.010 | 0.677 ± 0.009 |
|  | GraphCL | 0.435 ± 0.005 | 0.313 ± 0.024 | 0.629 ± 0.006 | 0.804 ± 0.014 | 0.675 ± 0.026 | 0.653 ± 0.012 | 0.737 ± 0.007 | 0.703 ± 0.016 |
|  | GRACE | 0.579 ± 0.007 | 0.339 ± 0.009 | 0.681 ± 0.006 | 0.838 ± 0.008 | 0.725 ± 0.014 | 0.678 ± 0.010 | 0.753 ± 0.010 | 0.712 ± 0.014 |
|  | G2P2 | 0.578 ± 0.007 | 0.360 ± 0.022 | 0.711 ± 0.007 | 0.838 ± 0.010 | 0.748 ± 0.009 | 0.710 ± 0.008 | 0.758 ± 0.009 | 0.725 ± 0.010 |
|  | TAGA | **0.631 ± 0.008** | 0.375 ± 0.021 | **0.731 ± 0.006** | 0.849 ± 0.008 | **0.754 ± 0.022** | **0.738 ± 0.015** | **0.787 ± 0.007** | **0.802 ± 0.014** |
|  | TAGA-rw | 0.595 ± 0.010 | **0.385 ± 0.016** | 0.704 ± 0.010 | **0.853 ± 0.005** | 0.749 ± 0.023 | 0.716 ± 0.010 | 0.776 ± 0.011 | 0.767 ± 0.021 |

Table 4: Full table of performance in zero-shot and few-shot node classification for each dataset and setting. The best-performing model is highlighted in **bold**, and the second-best performing model is underlined.

### B.5 ADDITIONAL ABLATION STUDIES

Here we have included an ablation analysis to verify the effectiveness of neighborhood size. The results in Table 8 demonstrate that our method achieves stable performance when using a neighborhood size of 2 or more orders.

## C ADDITIONAL TECHNICAL DETAILS

**Efficiency Comparison with Directly Using PLM Embeddings.** It is worth noting that the textual embeddings of *TofG* views $\mathbf{h}(v_i)$ can directly represent the entire TAG. However, it may cause significant scalability and efficiency issue during the inference phase. Existing PLMs typically adopts transformer architecture and it has a quadratic complexity with the input number of text tokens, this is especially important to TAGs since the number of input size grows exponentially with the number of neighborhood hops. By aligning the knowledge from PLM with GNN model through our framework, we can simultaneously maintain generalization ability of TAG embeddings and high efficiency and scalability to large-sized graphs.

| Text Encoder | Model | arxiv | children | computers | cora | history | photo | pubmed | sports |
|---|---|---|---|---|---|---|---|---|---|
| UAE-Large-V1 | PLM | $0.500 \pm 0.001$ | $0.094 \pm 0.003$ | $0.427 \pm 0.001$ | $0.624 \pm 0.005$ | $0.169 \pm 0.001$ | $0.387 \pm 0.009$ | $0.475 \pm 0.008$ | $0.316 \pm 0.002$ |
| | GraphMAE | $0.104 \pm 0.001$ | $0.021 \pm 0.001$ | $0.049 \pm 0.001$ | $0.194 \pm 0.006$ | $0.019 \pm 0.001$ | $0.152 \pm 0.001$ | $0.438 \pm 0.001$ | $0.112 \pm 0.001$ |
| | GraphCL | $0.089 \pm 0.001$ | $0.037 \pm 0.001$ | $0.173 \pm 0.001$ | $0.176 \pm 0.003$ | $0.191 \pm 0.001$ | $0.174 \pm 0.001$ | $0.368 \pm 0.001$ | $0.140 \pm 0.001$ |
| | GRACE | $0.045 \pm 0.001$ | $0.034 \pm 0.001$ | $0.169 \pm 0.001$ | $0.146 \pm 0.004$ | $0.079 \pm 0.001$ | $0.025 \pm 0.001$ | $0.335 \pm 0.001$ | $0.057 \pm 0.001$ |
| | G2P2 | $0.453 \pm 0.002$ | $0.201 \pm 0.001$ | $0.453 \pm 0.001$ | $0.644 \pm 0.004$ | $0.322 \pm 0.003$ | $0.452 \pm 0.001$ | $0.576 \pm 0.006$ | $0.436 \pm 0.001$ |
| | TAGA | $0.537 \pm 0.003$ | $0.224 \pm 0.001$ | $0.498 \pm 0.004$ | $0.682 \pm 0.005$ | $0.351 \pm 0.001$ | $0.419 \pm 0.001$ | $0.616 \pm 0.009$ | $0.448 \pm 0.003$ |
| Text-embedding-3-small | PLM | $0.351 \pm 0.001$ | $0.098 \pm 0.002$ | $0.434 \pm 0.005$ | $0.561 \pm 0.006$ | $0.125 \pm 0.001$ | $0.321 \pm 0.001$ | $0.306 \pm 0.001$ | $0.424 \pm 0.002$ |
| | GraphMAE | $0.101 \pm 0.001$ | $0.025 \pm 0.001$ | $0.108 \pm 0.001$ | $0.162 \pm 0.003$ | $0.158 \pm 0.001$ | $0.033 \pm 0.001$ | $0.205 \pm 0.001$ | $0.364 \pm 0.001$ |
| | GraphCL | $0.127 \pm 0.001$ | $0.045 \pm 0.001$ | $0.282 \pm 0.001$ | $0.197 \pm 0.004$ | $0.106 \pm 0.001$ | $0.163 \pm 0.001$ | $0.383 \pm 0.001$ | $0.240 \pm 0.003$ |
| | GRACE | $0.023 \pm 0.001$ | $0.022 \pm 0.001$ | $0.117 \pm 0.001$ | $0.085 \pm 0.004$ | $0.039 \pm 0.001$ | $0.037 \pm 0.001$ | $0.319 \pm 0.001$ | $0.088 \pm 0.001$ |
| | G2P2 | $0.332 \pm 0.001$ | $0.092 \pm 0.001$ | $0.449 \pm 0.001$ | $0.637 \pm 0.006$ | $0.168 \pm 0.001$ | $0.298 \pm 0.001$ | $0.569 \pm 0.001$ | $0.511 \pm 0.003$ |
| | TAGA | $0.369 \pm 0.001$ | $0.084 \pm 0.001$ | $0.615 \pm 0.001$ | $0.668 \pm 0.005$ | $0.264 \pm 0.001$ | $0.423 \pm 0.001$ | $0.639 \pm 0.001$ | $0.548 \pm 0.003$ |

Table 5: Zero-shot node classification performance.

| $k$-Shot | Model | Arxiv | Children | Computers | Cora | History | Photo | Pubmed | Sports |
|---|---|---|---|---|---|---|---|---|---|
| 1 | PLM | $0.280 \pm 0.044$ | $0.122 \pm 0.042$ | $0.238 \pm 0.039$ | $0.412 \pm 0.080$ | $0.284 \pm 0.078$ | $0.230 \pm 0.051$ | $0.503 \pm 0.067$ | $0.282 \pm 0.068$ |
| | GraphMAE | $0.255 \pm 0.041$ | $0.128 \pm 0.028$ | $0.300 \pm 0.052$ | $0.474 \pm 0.058$ | $0.231 \pm 0.052$ | $0.304 \pm 0.066$ | $0.492 \pm 0.076$ | $0.270 \pm 0.042$ |
| | GRACE | $0.263 \pm 0.034$ | $0.138 \pm 0.035$ | $0.336 \pm 0.051$ | $0.435 \pm 0.071$ | $0.266 \pm 0.085$ | $0.295 \pm 0.053$ | $0.514 \pm 0.095$ | $0.282 \pm 0.045$ |
| | G2P2 | $0.308 \pm 0.052$ | $0.145 \pm 0.029$ | $0.359 \pm 0.044$ | $0.477 \pm 0.082$ | $0.361 \pm 0.092$ | $0.372 \pm 0.066$ | $0.522 \pm 0.085$ | $0.356 \pm 0.042$ |
| | TAGA | $0.323 \pm 0.040$ | $0.180 \pm 0.073$ | $0.380 \pm 0.062$ | $0.509 \pm 0.089$ | $0.413 \pm 0.114$ | $0.417 \pm 0.077$ | $0.563 \pm 0.062$ | $0.440 \pm 0.070$ |
| 3 | PLM | $0.436 \pm 0.036$ | $0.194 \pm 0.029$ | $0.318 \pm 0.038$ | $0.588 \pm 0.036$ | $0.448 \pm 0.071$ | $0.352 \pm 0.044$ | $0.611 \pm 0.051$ | $0.392 \pm 0.041$ |
| | GraphMAE | $0.379 \pm 0.039$ | $0.182 \pm 0.025$ | $0.389 \pm 0.035$ | $0.634 \pm 0.044$ | $0.362 \pm 0.050$ | $0.432 \pm 0.051$ | $0.597 \pm 0.061$ | $0.363 \pm 0.050$ |
| | GRACE | $0.398 \pm 0.031$ | $0.200 \pm 0.038$ | $0.442 \pm 0.045$ | $0.622 \pm 0.043$ | $0.404 \pm 0.057$ | $0.447 \pm 0.053$ | $0.620 \pm 0.055$ | $0.398 \pm 0.045$ |
| | G2P2 | $0.430 \pm 0.027$ | $0.207 \pm 0.038$ | $0.469 \pm 0.042$ | $0.623 \pm 0.033$ | $0.508 \pm 0.073$ | $0.528 \pm 0.049$ | $0.641 \pm 0.064$ | $0.464 \pm 0.050$ |
| | TAGA | $0.445 \pm 0.035$ | $0.241 \pm 0.062$ | $0.497 \pm 0.035$ | $0.695 \pm 0.050$ | $0.551 \pm 0.094$ | $0.551 \pm 0.045$ | $0.659 \pm 0.058$ | $0.586 \pm 0.057$ |
| 5 | PLM | $0.500 \pm 0.019$ | $0.210 \pm 0.025$ | $0.377 \pm 0.027$ | $0.641 \pm 0.031$ | $0.557 \pm 0.040$ | $0.420 \pm 0.037$ | $0.632 \pm 0.040$ | $0.478 \pm 0.056$ |
| | GraphMAE | $0.425 \pm 0.028$ | $0.212 \pm 0.029$ | $0.434 \pm 0.036$ | $0.704 \pm 0.038$ | $0.459 \pm 0.038$ | $0.489 \pm 0.038$ | $0.625 \pm 0.049$ | $0.452 \pm 0.040$ |
| | GRACE | $0.445 \pm 0.028$ | $0.227 \pm 0.031$ | $0.472 \pm 0.040$ | $0.685 \pm 0.027$ | $0.481 \pm 0.061$ | $0.515 \pm 0.042$ | $0.628 \pm 0.047$ | $0.482 \pm 0.045$ |
| | G2P2 | $0.466 \pm 0.025$ | $0.240 \pm 0.034$ | $0.510 \pm 0.039$ | $0.703 \pm 0.032$ | $0.617 \pm 0.053$ | $0.583 \pm 0.051$ | $0.640 \pm 0.051$ | $0.565 \pm 0.055$ |
| | TAGA | $0.483 \pm 0.022$ | $0.263 \pm 0.031$ | $0.543 \pm 0.038$ | $0.752 \pm 0.028$ | $0.636 \pm 0.046$ | $0.602 \pm 0.041$ | $0.649 \pm 0.044$ | $0.664 \pm 0.061$ |
| 10 | PLM | $0.526 \pm 0.013$ | $0.240 \pm 0.018$ | $0.463 \pm 0.029$ | $0.690 \pm 0.017$ | $0.639 \pm 0.038$ | $0.491 \pm 0.028$ | $0.679 \pm 0.023$ | $0.535 \pm 0.038$ |
| | GraphMAE | $0.461 \pm 0.017$ | $0.234 \pm 0.014$ | $0.511 \pm 0.028$ | $0.761 \pm 0.023$ | $0.535 \pm 0.042$ | $0.543 \pm 0.035$ | $0.659 \pm 0.028$ | $0.508 \pm 0.028$ |
| | GRACE | $0.488 \pm 0.018$ | $0.251 \pm 0.015$ | $0.552 \pm 0.028$ | $0.754 \pm 0.018$ | $0.567 \pm 0.054$ | $0.567 \pm 0.031$ | $0.670 \pm 0.025$ | $0.529 \pm 0.033$ |
| | G2P2 | $0.527 \pm 0.014$ | $0.269 \pm 0.018$ | $0.598 \pm 0.031$ | $0.753 \pm 0.020$ | $0.649 \pm 0.046$ | $0.632 \pm 0.037$ | $0.691 \pm 0.029$ | $0.618 \pm 0.037$ |
| | TAGA | $0.521 \pm 0.017$ | $0.288 \pm 0.025$ | $0.622 \pm 0.025$ | $0.788 \pm 0.021$ | $0.679 \pm 0.041$ | $0.651 \pm 0.048$ | $0.714 \pm 0.024$ | $0.705 \pm 0.045$ |
| 20 | PLM | $0.526 \pm 0.013$ | $0.240 \pm 0.018$ | $0.463 \pm 0.029$ | $0.690 \pm 0.017$ | $0.639 \pm 0.038$ | $0.491 \pm 0.028$ | $0.679 \pm 0.023$ | $0.535 \pm 0.038$ |
| | GraphMAE | $0.501 \pm 0.009$ | $0.264 \pm 0.013$ | $0.558 \pm 0.015$ | $0.801 \pm 0.014$ | $0.597 \pm 0.033$ | $0.596 \pm 0.016$ | $0.689 \pm 0.021$ | $0.572 \pm 0.025$ |
| | GRACE | $0.521 \pm 0.011$ | $0.277 \pm 0.013$ | $0.605 \pm 0.017$ | $0.791 \pm 0.017$ | $0.640 \pm 0.037$ | $0.615 \pm 0.02$ | $0.704 \pm 0.029$ | $0.607 \pm 0.027$ |
| | G2P2 | $0.556 \pm 0.010$ | $0.301 \pm 0.015$ | $0.649 \pm 0.015$ | $0.813 \pm 0.012$ | $0.716 \pm 0.025$ | $0.672 \pm 0.015$ | $0.726 \pm 0.025$ | $0.690 \pm 0.025$ |
| | TAGA | $0.561 \pm 0.010$ | $0.319 \pm 0.023$ | $0.673 \pm 0.014$ | $0.814 \pm 0.012$ | $0.721 \pm 0.035$ | $0.694 \pm 0.021$ | $0.745 \pm 0.022$ | $0.759 \pm 0.026$ |
| 50 | PLM | $0.526 \pm 0.013$ | $0.240 \pm 0.018$ | $0.463 \pm 0.029$ | $0.690 \pm 0.017$ | $0.639 \pm 0.038$ | $0.491 \pm 0.028$ | $0.679 \pm 0.023$ | $0.535 \pm 0.038$ |
| | GraphMAE | $0.541 \pm 0.007$ | $0.300 \pm 0.008$ | $0.612 \pm 0.015$ | $0.815 \pm 0.008$ | $0.657 \pm 0.012$ | $0.631 \pm 0.010$ | $0.729 \pm 0.011$ | $0.631 \pm 0.018$ |
| | GRACE | $0.553 \pm 0.007$ | $0.314 \pm 0.012$ | $0.649 \pm 0.012$ | $0.818 \pm 0.012$ | $0.706 \pm 0.017$ | $0.661 \pm 0.019$ | $0.732 \pm 0.014$ | $0.678 \pm 0.022$ |
| | G2P2 | $0.578 \pm 0.009$ | $0.340 \pm 0.011$ | $0.692 \pm 0.012$ | $0.827 \pm 0.013$ | $0.738 \pm 0.009$ | $0.700 \pm 0.014$ | $0.758 \pm 0.009$ | $0.725 \pm 0.014$ |
| | TAGA | $0.586 \pm 0.010$ | $0.348 \pm 0.015$ | $0.712 \pm 0.012$ | $0.836 \pm 0.010$ | $0.743 \pm 0.022$ | $0.715 \pm 0.016$ | $0.771 \pm 0.011$ | $0.784 \pm 0.016$ |
| 100 | PLM | $0.592 \pm 0.005$ | $0.337 \pm 0.013$ | $0.610 \pm 0.008$ | $0.753 \pm 0.014$ | $0.753 \pm 0.008$ | $0.634 \pm 0.015$ | $0.771 \pm 0.005$ | $0.690 \pm 0.013$ |
| | GraphMAE | $0.573 \pm 0.005$ | $0.319 \pm 0.008$ | $0.650 \pm 0.008$ | $0.835 \pm 0.007$ | $0.684 \pm 0.011$ | $0.655 \pm 0.012$ | $0.744 \pm 0.010$ | $0.677 \pm 0.009$ |
| | GRACE | $0.579 \pm 0.007$ | $0.339 \pm 0.009$ | $0.681 \pm 0.006$ | $0.838 \pm 0.008$ | $0.725 \pm 0.014$ | $0.678 \pm 0.010$ | $0.753 \pm 0.010$ | $0.712 \pm 0.014$ |
| | G2P2 | $0.578 \pm 0.007$ | $0.360 \pm 0.009$ | $0.711 \pm 0.007$ | $0.838 \pm 0.010$ | $0.748 \pm 0.009$ | $0.710 \pm 0.008$ | $0.758 \pm 0.009$ | $0.725 \pm 0.010$ |
| | TAGA | $0.631 \pm 0.008$ | $0.375 \pm 0.021$ | $0.731 \pm 0.006$ | $0.849 \pm 0.008$ | $0.754 \pm 0.022$ | $0.738 \pm 0.015$ | $0.787 \pm 0.007$ | $0.802 \pm 0.014$ |

Table 6: Performance of all few-shot node classification for each dataset. The text encoder choice is `UAE-Large-V1`.

**Enabling Zero-Shot and Few-Shot Predictions.** Our pretrained strategy ensures that the embeddings obtained from the GNN models at each layer remain aligned within the textual embedding space. This alignment enables direct zero-shot predictions using the self-supervised trained embeddings without requiring any additional fine-tuning.

Specifically, suppose there are $L$ prediction labels $\{l_1, l_2, \ldots, l_L\}$. Their textual embeddings are obtained through the pretrained language model (PLM) as follows:

$$h^{(l)}(l_i) = \text{PLM}(l_i) \quad \text{for } i \in \{1, \ldots, L\} \tag{6}$$

| $k$-Shot | Model | Arxiv | Children | Computers | Cora | History | Photo | Pubmed | Sports |
|---|---|---|---|---|---|---|---|---|---|
| 1 | PLM | $0.199 \pm 0.044$ | $0.106 \pm 0.025$ | $0.347 \pm 0.084$ | $0.486 \pm 0.095$ | $0.285 \pm 0.108$ | $0.339 \pm 0.055$ | $0.491 \pm 0.066$ | $0.443 \pm 0.098$ |
| | GraphMAE | $0.167 \pm 0.041$ | $0.112 \pm 0.052$ | $0.257 \pm 0.037$ | $0.447 \pm 0.095$ | $0.268 \pm 0.063$ | $0.263 \pm 0.080$ | $0.456 \pm 0.069$ | $0.331 \pm 0.090$ |
| | GRACE | $0.224 \pm 0.038$ | $0.136 \pm 0.034$ | $0.329 \pm 0.046$ | $0.403 \pm 0.067$ | $0.304 \pm 0.096$ | $0.312 \pm 0.049$ | $0.513 \pm 0.086$ | $0.287 \pm 0.039$ |
| | G2P2 | $0.308 \pm 0.052$ | $0.145 \pm 0.029$ | $0.359 \pm 0.044$ | $0.477 \pm 0.082$ | $0.361 \pm 0.092$ | $0.372 \pm 0.066$ | $0.522 \pm 0.085$ | $0.356 \pm 0.042$ |
| | TAGA | $0.306 \pm 0.057$ | $0.173 \pm 0.072$ | $0.430 \pm 0.067$ | $0.523 \pm 0.101$ | $0.395 \pm 0.101$ | $0.431 \pm 0.083$ | $0.581 \pm 0.073$ | $0.510 \pm 0.099$ |
| 3 | PLM | $0.322 \pm 0.046$ | $0.148 \pm 0.024$ | $0.495 \pm 0.061$ | $0.66 \pm 0.037$ | $0.422 \pm 0.075$ | $0.438 \pm 0.044$ | $0.608 \pm 0.033$ | $0.577 \pm 0.082$ |
| | GraphMAE | $0.276 \pm 0.033$ | $0.169 \pm 0.051$ | $0.339 \pm 0.038$ | $0.657 \pm 0.038$ | $0.425 \pm 0.097$ | $0.347 \pm 0.048$ | $0.553 \pm 0.060$ | $0.398 \pm 0.064$ |
| | GRACE | $0.360 \pm 0.030$ | $0.191 \pm 0.037$ | $0.455 \pm 0.045$ | $0.580 \pm 0.041$ | $0.448 \pm 0.067$ | $0.461 \pm 0.045$ | $0.623 \pm 0.064$ | $0.426 \pm 0.045$ |
| | G2P2 | $0.430 \pm 0.027$ | $0.207 \pm 0.038$ | $0.469 \pm 0.042$ | $0.623 \pm 0.033$ | $0.508 \pm 0.073$ | $0.528 \pm 0.049$ | $0.641 \pm 0.064$ | $0.464 \pm 0.050$ |
| | TAGA | $0.442 \pm 0.023$ | $0.248 \pm 0.052$ | $0.548 \pm 0.058$ | $0.702 \pm 0.032$ | $0.523 \pm 0.08$ | $0.575 \pm 0.047$ | $0.683 \pm 0.056$ | $0.67 \pm 0.062$ |
| 5 | PLM | $0.365 \pm 0.037$ | $0.174 \pm 0.039$ | $0.55 \pm 0.036$ | $0.705 \pm 0.02$ | $0.522 \pm 0.094$ | $0.502 \pm 0.039$ | $0.601 \pm 0.032$ | $0.67 \pm 0.05$ |
| | GraphMAE | $0.308 \pm 0.030$ | $0.196 \pm 0.059$ | $0.384 \pm 0.026$ | $0.711 \pm 0.030$ | $0.511 \pm 0.058$ | $0.412 \pm 0.032$ | $0.563 \pm 0.068$ | $0.484 \pm 0.038$ |
| | GRACE | $0.399 \pm 0.026$ | $0.223 \pm 0.028$ | $0.501 \pm 0.043$ | $0.635 \pm 0.028$ | $0.513 \pm 0.051$ | $0.527 \pm 0.040$ | $0.640 \pm 0.052$ | $0.521 \pm 0.049$ |
| | G2P2 | $0.466 \pm 0.025$ | $0.240 \pm 0.034$ | $0.510 \pm 0.039$ | $0.703 \pm 0.032$ | $0.617 \pm 0.053$ | $0.583 \pm 0.051$ | $0.640 \pm 0.051$ | $0.565 \pm 0.055$ |
| | TAGA | $0.468 \pm 0.023$ | $0.299 \pm 0.034$ | $0.584 \pm 0.04$ | $0.74 \pm 0.031$ | $0.618 \pm 0.067$ | $0.6 \pm 0.041$ | $0.676 \pm 0.048$ | $0.735 \pm 0.063$ |
| 10 | PLM | $0.398 \pm 0.024$ | $0.189 \pm 0.026$ | $0.627 \pm 0.025$ | $0.741 \pm 0.018$ | $0.586 \pm 0.056$ | $0.541 \pm 0.022$ | $0.667 \pm 0.025$ | $0.708 \pm 0.039$ |
| | GraphMAE | $0.375 \pm 0.017$ | $0.208 \pm 0.011$ | $0.469 \pm 0.029$ | $0.763 \pm 0.027$ | $0.564 \pm 0.047$ | $0.491 \pm 0.034$ | $0.613 \pm 0.034$ | $0.539 \pm 0.046$ |
| | GRACE | $0.449 \pm 0.018$ | $0.249 \pm 0.019$ | $0.577 \pm 0.027$ | $0.714 \pm 0.023$ | $0.601 \pm 0.047$ | $0.578 \pm 0.030$ | $0.682 \pm 0.025$ | $0.569 \pm 0.039$ |
| | G2P2 | $0.527 \pm 0.014$ | $0.269 \pm 0.018$ | $0.598 \pm 0.031$ | $0.753 \pm 0.020$ | $0.649 \pm 0.046$ | $0.632 \pm 0.037$ | $0.691 \pm 0.029$ | $0.618 \pm 0.037$ |
| | TAGA | $0.509 \pm 0.020$ | $0.315 \pm 0.028$ | $0.661 \pm 0.028$ | $0.781 \pm 0.018$ | $0.67 \pm 0.049$ | $0.646 \pm 0.033$ | $0.724 \pm 0.022$ | $0.756 \pm 0.032$ |
| 20 | PLM | $0.434 \pm 0.016$ | $0.223 \pm 0.032$ | $0.659 \pm 0.014$ | $0.767 \pm 0.015$ | $0.641 \pm 0.04$ | $0.581 \pm 0.015$ | $0.712 \pm 0.021$ | $0.761 \pm 0.026$ |
| | GraphMAE | $0.429 \pm 0.011$ | $0.236 \pm 0.020$ | $0.535 \pm 0.023$ | $0.799 \pm 0.014$ | $0.625 \pm 0.024$ | $0.559 \pm 0.017$ | $0.655 \pm 0.030$ | $0.602 \pm 0.028$ |
| | GRACE | $0.486 \pm 0.014$ | $0.282 \pm 0.015$ | $0.613 \pm 0.019$ | $0.770 \pm 0.017$ | $0.654 \pm 0.027$ | $0.629 \pm 0.016$ | $0.697 \pm 0.022$ | $0.657 \pm 0.025$ |
| | G2P2 | $0.556 \pm 0.010$ | $0.301 \pm 0.015$ | $0.649 \pm 0.015$ | $0.813 \pm 0.012$ | $0.716 \pm 0.025$ | $0.672 \pm 0.015$ | $0.726 \pm 0.025$ | $0.690 \pm 0.025$ |
| | TAGA | $0.547 \pm 0.010$ | $0.332 \pm 0.023$ | $0.691 \pm 0.017$ | $0.805 \pm 0.011$ | $0.708 \pm 0.039$ | $0.682 \pm 0.015$ | $0.745 \pm 0.027$ | $0.808 \pm 0.022$ |
| 50 | PLM | $0.480 \pm 0.007$ | $0.252 \pm 0.022$ | $0.695 \pm 0.010$ | $0.785 \pm 0.009$ | $0.702 \pm 0.02$ | $0.609 \pm 0.013$ | $0.749 \pm 0.011$ | $0.784 \pm 0.014$ |
| | GraphMAE | $0.477 \pm 0.010$ | $0.278 \pm 0.012$ | $0.603 \pm 0.012$ | $0.819 \pm 0.011$ | $0.675 \pm 0.019$ | $0.630 \pm 0.015$ | $0.692 \pm 0.016$ | $0.673 \pm 0.021$ |
| | GRACE | $0.520 \pm 0.006$ | $0.324 \pm 0.012$ | $0.664 \pm 0.013$ | $0.806 \pm 0.014$ | $0.694 \pm 0.022$ | $0.668 \pm 0.020$ | $0.727 \pm 0.015$ | $0.712 \pm 0.020$ |
| | G2P2 | $0.578 \pm 0.009$ | $0.340 \pm 0.011$ | $0.692 \pm 0.012$ | $0.827 \pm 0.013$ | $0.738 \pm 0.009$ | $0.700 \pm 0.014$ | $0.758 \pm 0.009$ | $0.725 \pm 0.014$ |
| | TAGA | $0.576 \pm 0.009$ | $0.368 \pm 0.014$ | $0.734 \pm 0.007$ | $0.826 \pm 0.009$ | $0.738 \pm 0.021$ | $0.717 \pm 0.016$ | $0.773 \pm 0.009$ | $0.828 \pm 0.014$ |
| 100 | PLM | $0.508 \pm 0.005$ | $0.272 \pm 0.010$ | $0.722 \pm 0.007$ | $0.800 \pm 0.014$ | $0.73 \pm 0.015$ | $0.629 \pm 0.009$ | $0.772 \pm 0.008$ | $0.802 \pm 0.006$ |
| | GraphMAE | $0.499 \pm 0.008$ | $0.298 \pm 0.014$ | $0.634 \pm 0.008$ | $0.844 \pm 0.010$ | $0.704 \pm 0.015$ | $0.652 \pm 0.017$ | $0.721 \pm 0.007$ | $0.709 \pm 0.011$ |
| | GRACE | $0.546 \pm 0.007$ | $0.344 \pm 0.008$ | $0.693 \pm 0.006$ | $0.823 \pm 0.013$ | $0.714 \pm 0.011$ | $0.688 \pm 0.011$ | $0.745 \pm 0.006$ | $0.753 \pm 0.010$ |
| | G2P2 | $0.578 \pm 0.007$ | $0.360 \pm 0.009$ | $0.711 \pm 0.007$ | $0.838 \pm 0.010$ | $0.748 \pm 0.009$ | $0.710 \pm 0.008$ | $0.758 \pm 0.009$ | $0.725 \pm 0.010$ |
| | TAGA | $0.602 \pm 0.007$ | $0.400 \pm 0.017$ | $0.747 \pm 0.009$ | $0.838 \pm 0.009$ | $0.755 \pm 0.017$ | $0.738 \pm 0.010$ | $0.786 \pm 0.006$ | $0.846 \pm 0.013$ |

Table 7: Performance of all few-shot node classification for each dataset. The text encoder choice is `Text-embedding-3-small`.

| Method | arxiv | children | computers | cora | history | photo | pubmed | sports |
|---|---|---|---|---|---|---|---|---|
| 3-order | 0.532 | 0.223 | 0.493 | 0.678 | 0.351 | 0.415 | 0.622 | 0.387 |
| 2-order | 0.537 | 0.224 | 0.498 | 0.682 | 0.344 | 0.419 | 0.616 | 0.408 |
| 1-order | 0.500 | 0.197 | 0.463 | 0.635 | 0.318 | 0.392 | 0.566 | 0.448 |
| Glo-GofT | 0.533 | 0.205 | 0.482 | 0.657 | 0.329 | 0.407 | 0.522 | 0.417 |

Table 8: Additional ablation studies results of zero-shot settings. Here we show the results with different orders of alignment at 1, 2 and 3 order. We also show the results of a variant, *Glo-GofT*, which only aligns the GNN embeddings that aggregate individual node's text embeddings but removes all higher-order Graph-of-Text embeddings.

The probability that node $v_i$ belongs to class $l_j$ is computed in an unsupervised manner by measuring the cosine similarity (or another appropriate similarity measure) between the learned GNN embeddings $h^{(g)}(v_i)$ and the label textual embeddings $h^{(l)}(l_j)$:

$$p(v_i \rightarrow l_j) = \frac{e^{\rho(h^{(g)}(v_i), h^{(l)}(l_j))}}{\sum_{k=1}^{L} e^{\rho(h^{(g)}(v_i), h^{(l)}(l_k))}} \tag{7}$$

The final predicted class of node $v_i$ is determined as follows:

$$l(v_i) = \text{argmax}_j \, p(v_i \rightarrow l_j) \tag{8}$$

where $l(v_i)$ is the predicted class label for node $v_i$, determined by selecting the class $l$ that maximizes the similarity measure $\rho$ between the GNN embedding of the node $h^{(g)}(v_i)$ and each of the label embeddings $h^{(l)}(l_j)$.

Additionally, to further refine the learned embeddings, we introduce a learnable transformation function for few-shot learning adaptation:

$$h^{(g)}_{\text{adapted}}(v_i) = g(h^{(g)}(v_i), \mathcal{D}_{\text{support}}) \tag{9}$$

where $g$ represents a transformation function with learnable parameters (e.g., a multi-layer perceptron), and $\mathcal{D}_{\text{support}}$ denotes a set of support examples for few-shot learning. This adapted embedding $h^{(g)}_{\text{adapted}}$ is then utilized to compute the updated predictive probabilities:

$$p(v_i \rightarrow l_j) = \frac{e^{\rho(h^{(g)}_{\text{adapted}}(v_i), h^{(l)}(l_j))}}{\sum_{k=1}^{L} e^{\rho(h^{(g)}_{\text{adapted}}(v_i), h^{(l)}(l_k))}} \tag{10}$$

## D  LIMITATIONS

This work aims to pioneer unsupervised representation learning in the text-attributed graph research domain. Our approach demonstrates significant performance improvements over existing state-of-the-art methods in zero-shot and few-shot prediction tasks. However, we acknowledge certain limitations. While our work pushes the boundaries of graph foundation models, the model's transfer capabilities may be limited when training and inference domains are vastly different (e.g., from social networks to chemical networks). We consider the development of a universal graph foundation model, capable of generalizing across diverse domains, to be an important direction for future research.

