# OpenReview forum: "TAGA: Text-Attributed Graph Self-Supervised Learning by Synergizing Graph and Text Mutual Transformations"
_ICLR.cc/2025/Conference — Submitted to ICLR 2025_

### Official Review · Reviewer_LcJB · 2024-10-18

**Soundness:** 2
**Presentation:** 3
**Contribution:** 2
**Rating:** 3
**Confidence:** 3

**Summary:**

The paper presents TAGA, a novel self-supervised learning framework designed to enhance representation learning on Text-Attributed Graphs (TAGs) by integrating both textual and structural information. By aligning the representation of Text-of-Graph and Graph-of-Text, TAGA effectively captures joint textual and structural information. Additionally, a structure-preserving random walk algorithm is introduced to facilitate efficient training on large TAGs.

**Strengths:**

- The writing is fluent and easily comprehensible.
- The authors focus on the issue of the lack of training labels in real-world scenarios and propose a self-supervised learning framework. They also emphasize leveraging the semantic relationships between node texts.
- A random walk method inspired by human reading patterns is proposed to reduce the time consumption of the proposed TAGA.

**Weaknesses:**

- I believe that how to construct the Text-of-Graph view for each node is a key issue in this work. I can understand its construction method from Algorithm 1. However, I think the paper still lacks details, as it needs to demonstrate the original topological structure through the text. I hope the authors can provide an example to clarify this point.
- I think the authors need to elaborate more on why it is necessary to align the Graph-of-Text views of different orders (as in Figure 2(a))

**Questions:**

1. There is a typo in the caption of Figure 2 "The Graph2Text module that transforms a Graph-of-Text view into a Graph-of-Text
view".
2. It would be more appropriate to place the table's title at the top.
3. In Tables 1 and 2, it can be observed that the performance of TAGA-rw, which uses the random walk method, is often better than when this method is not used. This is counterintuitive; could the authors provide an explanation for this?
4. Regarding the loss function, it seems that the number of positive and negative samples is not balanced. Should a hyperparameter be introduced to balance the losses of L_{positive} and L_{negative} ?

---

### Official Review · Reviewer_fJ4H · 2024-11-04

**Soundness:** 3
**Presentation:** 4
**Contribution:** 3
**Rating:** 6
**Confidence:** 5

**Summary:**

This paper focuses on self-supervised learning on text-attributed graphs (TAGs) and introduces TAGA. Unlike prior efforts, TAGA employs a graph2Text scheme to transform a TAG into text documentation, enabling the direct use of pre-trained language models (PLMs) for graph encoding. Furthermore, layer-wise contrastive learning is applied using a hierarchical text structure extracted by graph2Text. Extensive experiments are conducted to demonstrate the effectiveness of TAGA compared to baseline methods across various evaluation scenarios.

**Strengths:**

- The proposed graph2Text scheme is novel and offers a general pipeline for transforming TAGs into text documents.
- A structure-preserving random walk strategy is introduced to further improve training efficiency on large graphs.
- Extensive experiments across various evaluation scenarios demonstrate the effectiveness of TAGA compared to leading baseline methods.

**Weaknesses:**

- The evaluation setup is incomplete. Graph self-supervised learning is a well-established research area in the graph field. The standard evaluation setup is linear probing under semi-supervised or supervised settings. While the proposed model performs well in few-shot learning scenarios, it would be beneficial to compare all self-supervised graph models under standard semi-supervised settings, especially for datasets like Cora and PubMed, which were originally designed for semi-supervised learning. Please follow the standard semi-supervised splits in GCN paper, and compare against relevant baselines using this setup.
- More key related works should be included for comparison or at least discussed. For instance, GIANT [1], UniGraph [2], and UniGLM [3] are leading self-supervised learning methods on TAGs and should be compared. Additionally, works like [4] and [5] should be discussed, as they also integrate textual information and graph topology in a unified manner and support zero-shot and few-shot scenarios. Without a thorough discussion of these recent advances, it is challenging to assess the true capabilities of the graph2Text approach. Therefore, a dedicated subsection in the Related Work section is required to discuss these recent advances in self-supervised learning on TAGs, and explain how their approach compares or differs conceptually.
- Similarly, several graph pre-training methods should be considered for comparison or at least discussed, as they represent distinct directions in graph pre-training. Examples include MaskGAE[6] and S2GAE[7], which serve as edge-level graph-masked autoencoders.
- The experimental setup for baseline methods needs further clarification. Since many graph pre-training methods (e.g., GraphMAE, GraphCL, and GRACE) were not originally designed for TAGs, data preprocessing is required to encode textual attributes into node features. However, as shown in [5], the choice of PLM encoder can significantly impact the performance of these methods. For a fair comparison, you can add a paragraph in the experimental setup section detailing how to preprocess the textual attributes for the baseline methods. Additionally, conducting more experiments using different PLM backbones for the baselines and report the results.
- Given TAGA’s capability to train on large graphs, it would be interesting to examine its performance on TAG datasets of varying scales. For instance, would training TAGA on multiple TAGs rather than a single TAG graph further enhance its performance?

[1] Node feature extraction by selfsupervised multi-scale neighborhood prediction.

[2] UniGraph: Learning a Unified Cross-Domain Foundation Model for Text-Attributed Graphs.

[3] UniGLM: Training One Unified Language Model for Text-Attributed Graphs.

[4] Zerog: Investigating cross-dataset zero-shot transferability in graphs

[5] GAugLLM: Improving Graph Contrastive Learning for Text-attributed Graphs with Large Language Models

[6] Maskgae: Masked graph modeling meets graph autoencoders.

[7] S2GAE: Self-Supervised Graph Autoencoders Are Generalizable Learners with Graph Masking.

**Questions:**

The research problem addressed in this work is highly significant to the graph field, and the graph2Text concept is innovative. However, the experimental setup and choice of competitive methods could be enhanced by incorporating more advanced baselines and a standardized graph pre-training evaluation framework. For further details, please refer to the Weaknesses section.

---

### Official Review · Reviewer_8Vri · 2024-11-05

**Soundness:** 2
**Presentation:** 2
**Contribution:** 3
**Rating:** 5
**Confidence:** 3

**Summary:**

The paper introduces TAGA, a self-supervised framework for Text-Attributed Graphs (TAGs) that hierarchically  integrates structural and semantic information. TAGA captures joint insights effectively, and a new random walk algorithm ensures efficient training on large TAGs, demonstrating strong performance in zero-shot and few-shot scenarios.

**Strengths:**

The idea of hierarchically aligning graph and text is interesting.

**Weaknesses:**

1.**Unclear Contributions**: The contributions are not clearly articulated. While the idea of aligning hierarchical textual and graph knowledge is compelling, simply stating that the work *"constructs two complementary views"* does not convey its novelty. Although the authors introduce related works on integrating text and graphs in pre-training and adaptation paradigms [1,2,3,4] in Section 2, the differences and unique contributions of this work should be highlighted in Section 1, as these distinctions form the primary contributions of this work.\
2. **Missing Related Works and Baselines**: For downstream tasks with limited labeled data, even in the absence of textual data, graph prompt tuning methods [5,6,7,8] have demonstrated strong performance and should be discussed and used as baselines for comparison.\
3. **Lack of Theoretical Proof for TAGA**: The paper lacks a theoretical proof for the effectiveness of TAGA, which would strengthen the validity of the proposed approach.

[1] Wen et al. Augmenting low-resource text classification with graph-grounded pre-training and prompting. SIGIR 2023\
[2] Wen et al. "Prompt tuning on graph-augmented low-resource text classification." TKDE 2024\
[3] Liu et al. "One for all: Towards training one graph model for all classification tasks." ICLR 2024\
[4] Tang et al. Graphgpt: Graph instruction tuning for large language models. SIGIR 2024\
[5] Yu et al. "A Survey of Few-Shot Learning on Graphs: from Meta-Learning to Pre-Training and Prompt Learning." arXiv 2024\
[6] Liu et al. "Graphprompt: Unifying pre-training and downstream tasks for graph neural networks." WWW 2023\
[7] Yu et al. "Generalized graph prompt: Toward a unification of pre-training and downstream tasks on graphs." TKDE 2024\
[8] Fang et al. "Universal prompt tuning for graph neural networks."  NeurIPS 2024.

**Questions:**

I am open to adjust my score based on the authors rebuttal.

---

### Official Review · Reviewer_uMnw · 2024-11-05

**Soundness:** 2
**Presentation:** 2
**Contribution:** 2
**Rating:** 5
**Confidence:** 4

**Summary:**

This paper introduces TAGA, a self-supervised learning framework for Text-Attributed Graphs (TAGs) that aligns two complementary views: Text-of-Graph, which organizes node texts into a structured hierarchical document, and Graph-of-Text, which represents nodes and connections as graph data. The framework includes a Graph2Text method for structure-preserving document generation and a random walk algorithm for efficient training. Experiments on eight real-world datasets demonstrate TAGA's superior performance in zero-shot and few-shot learning tasks across different domains.

**Strengths:**

1. The paper presents a novel dual-view framework that represents TAGs as both hierarchical documents and graph structures, marking a significant departure from traditional single-view approaches in graph representation learning.
2. The extensive evaluation demonstrates consistent performance improvements over six state-of-the-art methods across eight diverse datasets, with comprehensive ablation studies and strong transfer learning results validating the method's robustness.
3. The work addresses the fundamental challenge of jointly modeling text and structure in graph representation learning, offering a practical solution that reduces dependency on labeled data and shows strong performance on real-world applications.

**Weaknesses:**

While the paper presents a novel and valuable contribution to TAG representation learning, its innovation trends towards increasing complexity rather than seeking elegant solutions. The insufficient analysis of its sophisticated components leaves readers with an understanding of "what" but limited insights into "why", making it difficult to fully appreciate the fundamental principles behind its effectiveness：
1. The paper misses critical discussion and comparison with recent TAG self-supervised methods, particularly "UniGraph: Learning a Unified Cross-Domain Foundation Model for Text-Attributed Graphs". This omission raises concerns about the claimed novelty and makes it difficult to assess the true advancement over existing TAG pre-training approaches.
2.  The dual-view alignment architecture is unnecessarily restricted to self-supervised learning, while this structural design could potentially benefit from other learning paradigms. The paper fails to explore the architecture's potential beyond self-supervised learning.
3. The preprocessing steps for generating dual views (hierarchical documents and graph structures) incur substantial computational and storage costs, but the paper neither analyzes these overheads nor compares total computation time (preprocessing + training) with baseline methods.
4. The paper arbitrarily adopts BFS tree with cross-edge references as the document structuring method without any theoretical justification or experimental comparison with alternative organization strategies (like DFS, topological sorting, or community-based structuring), leaving a critical gap in understanding why this specific choice is optimal or even suitable for preserving graph structure in document format.
5. Despite claiming efficiency through random walk sampling, the paper lacks comprehensive experiments on large-scale graphs, missing analysis of how the random walk strategy affects model performance, and providing no comparison of end-to-end processing time with existing methods.

**Questions:**

Please refer to Weaknesses.

---

### Meta-Review · Area_Chair_pX55 · 2024-12-15

**Metareview:**

The paper proposes an interesting idea of aligning graph and text contents hierarchically, with extensive experiments.

However, the paper may not be ready for publication due to missing critical related work, requiring additional clarifications on the experimental setup, and lack of empirical demonstration on large-scale graphs, as raised by reviewers.

As the authors did not provide any response, it is unclear if these issues can be addressed satisfactory.

**Additional Comments On Reviewer Discussion:**

The authors did not provide rebuttals.

---

### Decision · Program_Chairs · 2025-01-22

Reject